# **QBOi El Niño Southern Oscillation experiments: Teleconnections of the QBO**

Naoe, Hiroaki<sup>1</sup>, Jorge L. García-Franco<sup>2</sup>, Chang-Hyun Park<sup>3</sup>, Mario Rodrigo<sup>4</sup>, Froila M. Palmeiro<sup>4,5</sup>, Federico Serva<sup>6</sup>, Masakazu Taguchi<sup>7</sup>, Kohei Yoshida<sup>1</sup>, James A. Anstey<sup>8</sup>, Javier García-Serrano<sup>4,9</sup>, Seok-Woo Son<sup>3</sup>, Yoshio Kawatani<sup>10</sup>, Neal Butchart<sup>11</sup>, Kevin Hamilton<sup>12</sup>, Chih-Chieh Chen<sup>13</sup>, Anne Glanville<sup>13</sup>, Tobias Kerzenmacher<sup>14</sup>, François Lott<sup>15</sup>, Clara Orbe<sup>16</sup>, Scott Osprey<sup>17</sup>, Mijeong Park<sup>13</sup>, Jadwiga H. Richter<sup>13</sup>, Stefan Versick<sup>14</sup>, Shingo Watanabe<sup>18</sup>

<sup>1</sup>Meteorological Research Institute (MRI), Tsukuba, 305-0052, Japan

<sup>2</sup>National School of Earth Sciences (Escuela Nacional de Ciencias de la Tierra), UNAM, CDMX, 04510, Mexico

O 3School of Earth and Environmental Sciences, Seoul National University, Seoul, 08826, Korea

<sup>4</sup>Group of Meteorology, Universitat de Barcelona, Barcelona, 08028, Spain

<sup>5</sup>CMCC Foundation - Euro-Mediterranean Center on Climate Change, Bologna, 40127, Italy

<sup>6</sup>Institute of Marine Sciences, National Research Council (CNR-ISMAR), 00133, Italy

<sup>7</sup>Department of Earth Science, Aichi University of Education, Kariya, 448-0001, Japan

8Canadian Centre for Climate Modelling and Analysis, Environment and Climate Change Canada, V8N 1V8, Canada

<sup>9</sup>Barcelona Supercomputing Center (BSC), Barcelona, 08034, Spain

<sup>10</sup>Faculty of Environmental Earth Science, Hokkaido University, Sapporo, 060-0810, Japan

<sup>11</sup>Met Office, Exeter, EX1 3PB, UK

<sup>12</sup>International Pacific Research Center (IPCC) University of Hawaii, Honolulu, 96822, USA

<sup>13</sup>U. S. National Science Foundation National Center for Atmospheric Research (NSF NCAR), Boulder, 80305, USA

<sup>14</sup>Karlsruher Institut für Technologie (KIT), Karlsruhe, 76131, Germany

<sup>15</sup>Laboratoire de Météorologie Dynamique (LMD), Ecole Normale Supérieure, Paris, 75231, France

<sup>16</sup>National Aeronautics and Space Administration (NASA) Goddard Institute for Space Studies (GISS), New York, 10025, USA

<sup>17</sup>Atmospheric, Oceanic and Planetary Physics, University of Oxford, Oxford, OX1 3PU, UK

<sup>18</sup>Japan Agency for Marine-Earth Science and Technology (JAMSTEC), Yokohama, 236-0001, Japan

Correspondence to: Hiroaki Naoe (hnaoe@mri-jma.go.jp)

July, 2025

Revised, to be submitted to Weather and Climate Dynamics


Abstract. This study examines Quasi-Biennial Oscillation (QBO) teleconnections and their modulation by the El Niño-Southern Oscillation (ENSO), using a multi-model ensemble of the Atmospheric Processes And their Role in Climate (APARC) QBO initiative (QBOi) models. Some difficulties arise in examining observed QBO-ENSO teleconnections from distinguishing the QBO and ENSO influences outside of the QBO region, due to aliasing between the QBO and ENSO over

sea-surface temperatures corresponding to idealized El Niño or La Niña conditions (QBOi EN and LN experiments, respectively). In the Arctic winter climate, higher frequencies of sudden stratospheric warmings (SSWs) are found in EN than LN. The frequency differences in SSW between QBO westerly (QBO-W) and QBO easterly (QBO-E) are indistinguishable, suggesting that the polar vortex responses to the QBO are much weaker than those to the ENSO in these models. The Asia-Pacific subtropical jet (APJ) shifts significantly equatorward during QBO-W compared to QBO-E in observations, while the APJ-shift is not robust across models, regardless of the ENSO phases. In the tropics, these experiments do not show a robust or coherent QBO influence on precipitation. The sign and spatial pattern of the precipitation response vary widely across models and experiments, indicating that any potential QBO signal is strongly modulated by the prevailing phases of ENSO. The QBO teleconnection to the Walker circulation in boreal summer/autumn shows a consistent signal across observations and most models, with upper-level westerly and lower-level easterly anomalies over the Indian Ocean–Maritime Continent, although its amplitude and timing are model-dependent.

#### Short summary (500 characters, incl. spaces)

55

This study examines links between the stratospheric Quasi-Biennial Oscillation (QBO) and large-scale atmospheric circulations in the tropics, subtropics, and polar regions. The QBO teleconnections and their modulation by the El Niño–Southern Oscillation (ENSO) are investigated through a series of climate model experiments. While QBO teleconnections are qualitatively reproduced by the multi-model ensemble, they are not consistent due to modelled QBO bias and other systematic model biases.

Key words: stratosphere-troposphere coupling, teleconnection, QBO, ENSO

#### 1 Introduction

65

The Quasi-Biennial Oscillation (QBO) and the El Niño-Southern Oscillation (ENSO) are the leading modes of climate variability in the tropical stratosphere and tropical troposphere, respectively. The QBO is a semi-periodic wind variation characterized by downward propagating easterly and westerly wind regimes in the equatorial stratosphere with an average period of about 28 months (Baldwin et al., 2001; Anstey et al., 2022b). The QBO is an important source of predictability due to its long timescale and its teleconnections outside the tropical stratosphere. The QBO is primarily driven by vertical momentum fluxes due to upward-propagating equatorial wave activity generated by tropospheric convective systems (Lindzen and Holton, 1968; Holton and Lindzen, 1972; Plumb and McEwan, 1978).

Over the past a couple of decades, atmospheric general circulation models (GCMs) and Earth system models (ESMs) are being increasingly developed to include an internally generated QBO to represent more realistic modes of internal variability (e.g. Butchart et al., 2018). Most of these models require parameterization of unresolved gravity waves to simulate an internally generated QBO, including specific conditions of parameterized and/or resolved convection, high horizontal and vertical resolution, and weak implicit and explicit grid-scale dissipation (Anstey et al., 2022b). Although the QBO is primarily an equatorial stratospheric phenomenon, it impacts the climate system outside this region via teleconnections. We can obtain a more in-depth understanding of QBO teleconnections (extratropical impacts, tropical and subtropical impacts, and their interaction with other phenomena) by intercomparing many state-of-the-art, stratosphere-resolving models that simulate a QBO-like oscillation in the tropical stratosphere.

The QBO can influence the Northern Hemisphere (NH) winter stratosphere by modulating planetary-scale waves that distort the stratospheric polar vortex. The observed statistical relationship between the QBO phase and polar vortex strength is commonly referred to as the Holton-Tan effect (Holton and Tan, 1980, 1982). When the QBO in the lower stratosphere (~50 hPa) is in its westerly phase (QBO-W), the polar vortex is observed to be stronger and colder, and the likelihood of sudden stratospheric warming (SSW) events is reduced. Conversely, when the QBO is in its easterly phase (QBO-E), the stratospheric polar vortex is weaker, warmer, and more disturbed. The underlying mechanisms for this effect have been extensively examined by many observational and modeling studies. The mechanism proposed by Holton and Tan (1980) to explain this relationship involves a latitudinal shift of the zero-wind line, which acts as an effective waveguide for upward-propagating planetary waves in the NH winter stratosphere (Holton and Tan, 1980; Baldwin et al., 2001; Anstey and Shepherd 2014; Watson and Gray, 2014; Gray et al., 2018; Lu et al., 2020; Anstey et al., 2022b). A similar but distinct mechanism involves planetary waves interacting with the zonal wind anomalies associated with the QBO secondary circulation, not requiring zero-wind-line-induced wave breaking (Ruzmaikin et al., 2005; Naoe and Shibata, 2010; Garfinkel et al., 2012b; White et al., 2015; Naoe and Yoshida, 2019; Rao et al., 2020; Anstey et al., 2022b). A tropospheric pathway of the Holton-Tan relationship has also been proposed. This mechanism involves Rossby waves propagating from regions of tropical convection to higher

latitudes, including the Aleutian low-pressure region, and the stratospheric polar vortex is disturbed by the subsequent upward wave activity flux into the stratosphere, which is modulated through tropospheric processes (Yamazaki et al., 2020). Although the relative importance of these different mechanisms remains somewhat unclear, due to the QBO's long timescale these teleconnections may lead to increased predictability of the extratropical stratosphere on sub-seasonal time scales (Boer and Hamilton, 2008; Scaife et al., 2014; Garfinkel et al., 2018).

The QBO has also been suggested to affect the tropical troposphere by modifying deep convective activity and vertical wind shear along the tropopause (Gray et al., 1992; Collimore et al., 2003). The QBO-induced zonal-mean meridional circulation modulates the temperature vertical profile in the equatorial upper troposphere and lower stratosphere (UTLS), leading to a QBO signature in tropical tropopause temperature and wind. Although the idea of a "direct effect" of the QBO on the tropical and subtropical UTLS had been discussed in the literature since the 1960s, it was not yet widely accepted until the early 2000s (Hitchman et al., 2021). Recently a possible downward influence of the QBO on the tropical troposphere has been found in the Madden–Julian Oscillation (MJO) (Yoo and Son, 2016; Marshall et al., 2016; Son et al., 2017; Martin et al. 2021; Elsbury et al., in revision). For more recent reviews of stratosphere-troposphere coupling in the tropics, see Haynes et al. (2021) and Hitchman et al. (2021).






Observational and modeling studies suggest that the interannual variability of tropical precipitation is, at least partially, modulated by the phase of the QBO (Collimore et al., 2003; Liess and Geller, 2012; Gray et al., 2018). In observations, the OBO signal in tropical precipitation shows zonally asymmetric patterns, e.g. wetter conditions in the eastern Pacific Intertropical Convergence Zone (ITCZ) during QBO-W compared to QBO-E (Gray et al., 2018, Serva et al., 2022). The similarity between the QBO and ENSO signals in observations could potentially be caused by the higher number of El Niño events coinciding with QBO-W than with QBO-E (García-Franco et al., 2022). Serva et al. (2022) analyzed the simulated precipitation in Atmospheric Model Intercomparison Project (AMIP)-type simulations from the first phase of QBOi experiments (Butchart et al., 2018) and found that those simulations have limited ability to reproduce the observed modulation of the tropical tropopause level processes, even after subtracting the variability associated with the ENSO index. In these seasurface temperature (SST)-forced, free-running simulations, the east Pacific ITCZ precipitation response to the QBO, which resembles the observed pattern, is simulated by many, though not all models (Fig. 11 of Serva et al. (2022)). However, the simulated QBO signal on the tropopause is generally underestimated or not realistic. Also, Rao et al. (2020b) explored and evaluated three dynamical pathways (stratosphere polar vortex, North Pacific through the subtropical downward arching zonal wind, and tropical convection pathways) for impacts of the QBO on the troposphere, using the state-of-the-art CMIP5/6 models with a spontaneously generated QBO. They found that more than half of the models can reproduce at least one of the three pathways, but few models can reproduce all of the three routes. Using similar SST-forced, as well as ocean-atmosphere coupled simulations with a single model, García-Franco et al. (2023) suggested that the simulated precipitation response to the QBO is heavily dependent on the state of ENSO and the Walker circulation, the strength of the QBO and the ocean-atmosphere coupling.

In the subtropics, a direct influence of the QBO modulates the subtropical jet by the QBO secondary circulation. Observational studies indicate that the QBO can affect the subtropical jet variability especially in the Pacific sector (e.g. Garfinkel and Hartmann, 2011a; 2011b). During QBO-W, a horseshoe-shaped zonal wind anomaly forms in the upper troposphere and lower stratosphere associated with the equatorward shift of the Asian-Pacific jet (APJ) (Crooks and Gray, 2005; Simpson et al., 2009), and the resultant response is found even in the East Asian near the surface (Park et al., 2022; Park and Son, 2022). A study using QBO-resolving multi-model ensemble found no clear evidence of a QBO teleconnection to the subtropical Pacific-sector jet (Anstey et al., 2022c), while another multi-model study found that seven out of 17 models captured this effect (Rao et al., 2020b).







ENSO teleconnections to the NH winter stratosphere have been widely reported in a large number of observational studies (van Loon and Labitzke, 1987; Camp and Tung, 2007; Garfinkel and Hartmann, 2007; Song and Son, 2018) and in modeling studies (Taguchi and Hartmann, 2006; García-Serrano et al., 2017; Palmeiro et al., 2017, 2023; Trascasa-Castro et al., 2019; Weinberger et al., 2019). During El Niño winters, the polar vortex is weaker and the polar region is warmer than ENSO neutral years, while during strong La Niña winters, a weakening of the Aleutian low and destructive linear interference with the climatological wave pattern was identified (Iza et al., 2016). Observations showed that SSW events occur preferentially during both El Niño and La Niña winters than during ENSO-neutral winters (Butler and Polyani, 2011; Garfinkel et al., 2012a). However, there might be sampling errors due to the relatively short observational record (Domeisen et al., 2019), and increased SSWs during La Niña winters were sensitive to the SSW definition (Song and Son. 2018). Observed relationships between ENSO and SSWs were often not replicated by models. Models often simulated ENSO events and teleconnections that were considerably more linear compared to the available observational data (Domeisen et al., 2019). For example, there is no indication of any nonlinearities between EN and LN, while SSW frequencies for EN and LN are both similar, using a chemistry-climate model (Weinberger et al., 2019). Trascasa-Castro et al. (2019) investigated the effect of variations in ENSO amplitude on European winter climate with idealized SST anomalies, and they did not find evidence of a saturation of the stratospheric pathway due to strong El Nino forcing, as suggested in previous literature. Systematic model biases in atmospheric winds and temperatures would affect the ENSO-SSW connection (Tyrrell et al., 2022).

ENSO has significant impacts on the global atmospheric circulations, and QBO teleconnections may also be influenced by El Niño and La Niña. The QBO itself is affected by ENSO, with weaker QBO amplitude and faster QBO phase propagation under El Niño than La Niña conditions (Taguchi, 2010a). Previous studies that investigated the joint effects of QBO and ENSO on polar vortex variability in winter suggested that their interactions are nonlinear insofar as the Holton-Tan relationship is found to be significant in the La Niña phase but much weaker in the El Niño phase (Wei et al., 2007; Garfinkel and Hartmann, 2008; Calvo et al., 2009; Richter et al., 2011; Hansen et al., 2016). A recent observational study (Kumar et al., 2023) investigated the combined effects of the QBO and ENSO in modulating the extratropical winter troposphere during the 1979–2018 period. They found that during La Niña, QBO signals in the polar vortex were amplified and the polar vortex and subtropical jet were enhanced under QBO-W. During El Niño, a stronger subtropical jet and the warmer polar vortex were present under QBO-W. Ma et al. (2023) assessed the synergistic effects of QBO and ENSO on the North Atlantic winter

atmospheric circulation using model output and reanalysis data and found that the QBO and ENSO have a nonlinear combined effect on North Atlantic surface pressure anomalies, which arises because different pathways are preferred for different combinations of QBO and ENSO. In contrast, the polar vortex weakens more when El Niño and QBO easterly occur together than would be expected by the sum of their individual effects (Walsh et al., 2022). However, there remains a lack of consensus on the nature of nonlinearity in QBO–ENSO teleconnections in the extratropical circulation of the NH winter stratosphere.







In the tropical troposphere, the QBO and ENSO teleconnections remain less understood than those in the extratropics. A relatively small number of studies have analyzed tropical tropospheric QBO teleconnections using models that simulate the QBO (Rao et al., 2020; García-Franco et al., 2022, 2023; Serva et al., 2022). As noted by García-Franco et al. (2022, 2023), the observational record is likely too short to separate QBO teleconnections in the tropical troposphere from the strong influence of ENSO, because El Niño winters often coincide with the westerly phase of the OBO.

The goal of this study is to reexamine QBO teleconnections to the extratropics and tropics but now address combined QBO-ENSO influences using a new dataset of idealized ENSO experiments. Model experiments, which are capable of separating QBO and ENSO influences on the extratropical and tropical troposphere outside of the QBO region, are a valuable tool to study the modulation of QBO teleconnections by ENSO. To isolate the QBO teleconnections from the influence of ENSO, we conduct model integrations with annually-repeating prescribed SSTs characteristic of typical El Niño and La Niña conditions, removing ENSO diversity from consideration.

The Quasi-Biennial Oscillation initiative (QBOi), an international project supported by the World Climate Research Programme (WCRP) core project Atmospheric Processes And their Role in Climate (APARC), has focused on assessing internally generated QBOs in climate models and improving understanding of how to simulate a realistic QBO (Butchart et al., 2018; Anstey et al., 2022a,c; Bushell et al., 2022; Richter et al., 2022). In order to study QBO and ENSO teleconnections and their mutual interactions, QBOi has coordinated additional experiments building on the QBOi phase-1 experiments, referred here as the "QBOiENSO" experiments, using participating QBOi atmospheric general circulation models (AGCMs) and Earth System Models (ESMs) forced by prescribed "perpetual El Niño" and "perpetual La Niña" SSTs (Kawatani et al., in revision).

In this paper, we have examined QBO teleconnections modulated by ENSO and their robustness using this multi-model ensemble of QBO-resolving models that have run the QBOiENSO experiments, and evaluated them by comparison against the QBOi phase-1 "Experiment 2", which represents the control case of ENSO-neutral conditions. Further details of how the QBOiENSO experiments are constructed can be found in Kawatani et al. (in revision). The structure of the paper is as follows. Section 2 describes datasets of the QBOiENSO experiments and observations, and the analysis methods employed. Section 3 characterizes the combined effects of QBO-ENSO teleconnections on the polar winter stratosphere (Holton-Tan relationship). Sections 4 and 5 present the subtropical and tropical impacts of the QBO modified by ENSO, respectively. Finally, Section 6 provides a summary of our findings and discussion.

## 2 Data and Methods




We use nine AGCMs and ESMs participating in the QBOi project, conducting three experiments. The first one is the QBOi Experiment 2 using climatological SST and sea ice conditions (Butchart et al., 2018). We hereafter refer to it as the control (CTL) experiment. The other two experiments are the QBOiENSO experiments, QBOiElNino and QBOiLaNina (Kawatani et al., in revision). They are also time-slice experiments consistent with the QBOi Experiment 2 design, but prescribed "perpetual El Niño" and "perpetual La Niña" SSTs are used here. They are referred to hereafter as the EN and LN experiments, respectively. The models that performed the CTL, EN, and LN experiments are EC-EARTH3.3 (hereafter EC-EARTH for short), ECHAM5sh, EMAC, GISS-E2-2G (GISS for short), LMDz6 (LMDz for short), MIROC-AGCM-LL (MIROC-AGCM for short), MIROC-ESM, MRI-ESM2.0, and CESM1(WACCM5-110L) (WACCM for short). Their characteristics have been described in Butchart et al. (2018) and Kawatani et al. (in revision). MRI-ESM2.0 (Yukimoto et al., 2019) is an updated version of the model documented in Butchart et al. (2018), and it includes changes aimed at improving the modelled QBO (Naoe and Yoshida, 2019). Model integration years for three experiments are presented in Table 1. Due to data availability issues, EMAC is not included in Section 4 and 5.1.

Table 1. Model integration years

| Model name            | Years             |                        |
|-----------------------|-------------------|------------------------|
|                       | ¹QBOi Exp2        | <sup>2</sup> QBOi ENSO |
| <sup>3</sup> EC-EARTH | 101-yr            | 101-yr                 |
| <sup>4</sup> ECHAM5sh | 50-yr             | 40-yr                  |
| EMAC                  | 106-yr            | 106-yr                 |
| GISS-E2-2G            | $3 \times 30$ -yr | $3 \times 30$ -yr      |
| LMDz                  | 70-yr             | 82-yr                  |
| MIROC-AGCM            | $3 \times 30$ -yr | 100-yr                 |
| MIROC-ESM             | 3 × 100-yr        | 100-yr                 |
| MRI-ESM2.0            | 30-yr             | 50-yr                  |
| <sup>5</sup> WACCM    | 3 x 30-yr         | 100-yr                 |

<sup>&</sup>lt;sup>1</sup>QBOi Experiment 2 (or CTL experiment)

<sup>&</sup>lt;sup>2</sup>QBOi ENSO experiments (QBOiElNino and QBOiLaNina experiments)

<sup>&</sup>lt;sup>3</sup>EC-EARTH3.1 for QBOi Exp2 and EC-EARTH3.3 for QBOi ENSO

<sup>205 &</sup>lt;sup>4</sup>Only r2i1p1 is used in ECHAM5sh.

<sup>&</sup>lt;sup>5</sup>CESM1 (WACCM5-110L) is abbreviated to WACCM.

Observed teleconnections are quantified using a modern reanalysis dataset, the European Centre for Medium-Range Weather Forecasts (ECMWF) fifth generation atmospheric reanalysis (ERA5; Hersbach et al., 2020) in 1959–2021. The representation of the QBO in ERA5 as compared to other reanalyses is evaluated by Pahlavan et al. (2021) and Naoe et al. (2025). Observed precipitation is evaluated using the dataset of the Global Precipitation Climatology Project (GPCP; Adler et al., 2003, 2016) in 1979–2022. Because the design of QBOiENSO experiments used the Japan Meteorological Agency's (JMA) defined NINO.3 index (https://ds.data.jma.go.jp/tcc/tcc/products/elnino/index/index.html), the classification of ENSO phases is based on this index. We note that the QBOiENSO experiments are idealized, therefore we mostly rely on observation-based datasets to determine whether the model responses are at least qualitatively in agreement with the (short) observational record.








To determine if observed teleconnections are manifested in the model runs, models and observations are compared by applying the same OBO phase definitions to the models that are optimal for observed teleconnections. Here, we use 'standard' indices (e.g., 50-hPa equatorial wind for the QBO), without adjusting them on a model-by-model basis, for all analyses presented in Sections 3, 4 and 5.1. This can facilitate comparison with other works. As noted by Anstev et al. (2022c), different QBO indices can maximize the response of different teleconnections (e.g. Gray et al., 2018). Thus, making these choices can account for diversity of QBO signals (tropical convection, Walker circulation, subtropical jet response, extratropical basicstate zonal-mean flow for the Holton-Tan effect etc.), which may lead to variations in the diagnosed OBO teleconnections. Zonal wind biases need to be carefully considered when defining the QBO phases in model outputs, as noted by Serva et al. (2022). Here OBO phases are identified when the deseasonalized westerly and easterly zonal-mean zonal wind (OBO-W and QBO-E) averaged over 5° S-5° N (weighted by cosine of latitude) exceeds a given threshold value at selected pressure levels. Specifically, QBO-W and QBO-E are classified from December-January-February (DJF) zonal wind at 50 hPa using > 0.5 σ (standard deviation) and  $< -0.5 \,\sigma$  in Section 3.1 (Figs. 2 and 3), using  $> 0 \,\mathrm{m \, s^{-1}}$  and  $< 0 \,\mathrm{m \, s^{-1}}$  in Section 3.2 (Fig. 5), using  $> 0 \,\mathrm{m \, s^{-1}}$ 2 m s<sup>-1</sup> and  $\leq$  -2 m s<sup>-1</sup> in Section 5.1 (Figs. 8, 9, and 10), and from February-March zonal wind at 70 hPa using > 0.5  $\sigma$  and  $\leq$ -0.5 σ in Section 4 (Figs. 6 and 7). In Section 5.2, the strongest signal in each model is identified, considering model diversity and biases in the simulated QBOs and tropical convection, from May to November with QBO definitions provided in the legend of Figs. 11 and 12; the analysis is summarized in Fig. 13. This approach is used to highlight the model dependency and seasonality of the QBO signal on the Walker circulation. Using a common definition for QBO phases in terms of pressure level and season provides similar but weaker results (see Figs. S9, S10 and S11 using zonal wind at 70 hPa).

ENSO composites in observations are done in the extratropics and subtropics for individual seasons (Sections 3, 4, and 5.2) and in the tropics for individual months (Section 5.1). In Section 5.2, the Bonferroni correction, as described by Holm (1979), is used for the two-sided *t*-test when the QBO phase is not defined using the preferred 70 hPa level during June-July-August (JJA). In this method, the significance level of the statistical test is adjusted by dividing it by m, the number of tests performed, becoming more restrictive by increasing the confidence level. For instance, if the QBO definition is modified by season only, m = 2; if it is modified by both season and vertical level, m = 3. Accordingly,  $\alpha' = \alpha/m$ , where  $\alpha = 0.025$  (the 5% significance level for a two-sided test), and  $\alpha'$  denotes the adjusted threshold; implying that the corresponding p-value has to be smaller to reject the null hypothesis.

#### 3 QBO teleconnections: the extratropical route

A previous study about teleconnections of the QBO in a multi-model ensemble of QBO-resolving models (Anstey et al. 2022c) found that QBOi models underestimated the polar vortex response to the equatorial zonal wind at 50 hPa in comparison to reanalyses. They suggested that such weak responses were likely due to model errors, such as systematically weak QBO amplitudes near 50 hPa, affecting the teleconnection. Because most of the models that have run the QBOiElNino (EN) and QBOiLaNina (LN) experiments considered here are the same models whose QBOiExp2 (CTL) runs were analyzed by Anstey et al. (2022c), EN and LN runs may similarly underestimate the polar vortex response to the QBO. This section investigates the extratropical route of the QBO teleconnection modulated by ENSO. First, we examine the Holton-Tan effect, and then show the SSW statistics.

## 3.1 Holton-Tan relationship

Figure 1: Vertical profiles of correlation coefficient between the QBO zonal wind at 50 hPa averaged over 5° S-5° N and the polar-vortex zonal wind at 55°-65° N in December-January-February (DJF) for QBOi models and ERA5. Circles represent statistical significance at the 90 % level. Red and blue bars represent 5-95 % confidence ranges using a bootstrap method repeating 1000 times in EN and LN experiments for the models as well as El Niño and La Niña winters for ERA5. Number of winters available for each model run for each experiment (ENSO phase) are displayed at the upper panel. For example, 'NEU32EN15LN15' in the ERA5 panel means there are 32 ENSO-neutral, 15 El Niño, and 15 La Niña winters, respectively.

Figure 1 shows the correlation coefficient in DJF between the 50 hPa equatorial zonal wind at 5° S–5° N and the polar vortex strength at different altitudes in the CTL, EN, and LN experiments, together with ENSO-neutral, El Niño and La Niña winters for the ERA5 reanalysis. In the reanalysis, correlations are maximized over a fairly deep layer in the polar vortex, peaking 0.63 at 15 hPa during La Niña and 0.40 during El Niño. The correlation during the ENSO-neutral winter is slightly stronger than that of El Niño. The uncertainty range (horizontal bars) shows the 5–95% range of correlation coefficients derived from bootstrap resampling. Although the confidence interval for La Niña clearly excludes zero in the stratosphere, the confidence for El Niño is close to zero at many altitudes, demonstrating large uncertainty in the strength of the correlation especially for El Niño and ENSO-neutral winters.

Most of the model correlations show smaller uncertainty than ERA5 due to having larger sample sizes. Models (ECHAM5sh, EMAC, EC-EARTH, MIROC-ESM, MRI-ESM2.0, and WACCM) have positive correlation profiles in ENSO-neutral, albeit weak compared to reanalysis. Most models do not show a significant correlation in EN, and only four models (MRI-ESM2.0, ECHAM5sh, EMAC, and MIROC-AGCM) out of 9 reproduce observed positive correlations with confidence intervals excluding zero at some altitudes. It is noted in Fig. 2 of Kawatani et al. (in revision) from their simple, time-height cross-sections of the monthly and zonal-mean zonal winds over the equator in the EN and LN simulations that QBO in the ECHAM5sh for the EN experiment is irregular, with stalling in downward phases of easterlies and westerlies. They showed that the QBOs in GISS and LMDz for the LN experiment are more irregular, and westerly phases sometimes fail to propagate into the lower stratosphere. These results indicate that most models show weak positive correlations with the same sign as the reanalysis, but in most cases these correlations are not statistically significant. This means that inter-model differences in the QBO-polar vortex relationship, or differences between experiments for the same model, may not be distinguishable.

Figure 2 shows composite differences of zonal-mean zonal wind between QBO-W and QBO-E in the CTL, EN, and LN experiments. ERA5 clearly represents the Holton-Tan relationship under all three ENSO conditions (neutral, El Niño, and La Niña). The QBO teleconnection to the NH winter stratospheric polar vortex is the strongest in correlation with the amplitude of the QBO at 50 hPa (Anstey et al., 2022c). The vortex strength difference between QBO-W and QBO-E peaks at roughly 10 m s<sup>-1</sup> in the middle stratosphere (near 10 hPa) during DJF for the Neutral and El Niño groups, and the response is strongest in La Niña with a peak value of 15 m s<sup>-1</sup>. No model reproduces the observed-strength Holton-Tan relationship in all three experiments (CTL, EN and LN). Only two of the models reproduce observed responses within a half of the amplitude for the ENSO-neutral case (MRI-ESM2.0 and WACCM), and only the MRI-ESM2.0 also shows a stronger impact on the QBO on the vortex under the LN condition than under EN condition (however, that model has the wrong sign response for EN). In LN, four models (ECHAM5sh, GISS, MIROC-AGCM, and MRI-ESM2.0) are better at reproducing the observed response, peaking at a slight amplitude of ~3 m s<sup>-1</sup> in the polar vortex region. GISS shows a significant difference in EN, and a significant LN response just equatorward of 60° N.

Figure 2: Composite differences of DJF-mean zonal-mean zonal wind between QBO-W and QBO-E in the CTL, EN, and LN experiments including the ENSO neutral, El Niño, and La Niña winters for ERA5. QBO phases are classified using deseasonalized DJF zonal-mean zonal wind at 50 hPa averaged over  $5^{\circ}$  S- $5^{\circ}$  N using > 0.5  $\sigma$  for QBO-W and < -0.5  $\sigma$  for QBO-E. Contour interval is 3 m s<sup>-1</sup>. Dots represent statistical significance at the 90 % level. Number of winters available for each model run, and numbers of QBO-E and QBO-W winter classification are displayed at the upper right corner of each panel. For example, 'N100E30W41' in EC-EARTH and OBOiExp2 means there are 100 winters in which 30 OBO-E winters and 41 OBO-W winters are classified.

One may ask if a model-specific equatorial wind level such as 30 hPa (e.g., Rao et al. 2020a) can be more efficient for models to reproduce QBO's impact on the polar vortex (the Holton-Tan effect). We have examined a relationship of composite differences of zonal-mean zonal wind between polar vortex responses at 60° N and 10 hPa and QBO definition at 50 hPa (QBO50) and at 30 hPa (QBO30) (Fig. S1). Most models underestimate the equatorial QBO composite differences at 50 hPa compared to those at 30 hPa, and for some models the QBO is difficult to detect at 50 hPa; these results are similar to those

described in Rao et al. (2020a), which was on CMIP models. However, both panels (QBO50 and QBO30) show that most models underestimate equatorial QBOs and they are struggling to reproduce observed polar vortex responses to the QBO. We also have examined whether model performance of QBO amplitude and/or climatological polar night jet strength is related to the ability of model to capture the QBO-induced polar vortex responses (not shown), here hypothesizing that the HTR relationship (polar vortex) route of the QBO teleconnection could be manifested by these two factors. QBO amplitudes at 50 hPa for most models are poor performance, in agreement with previous studies (Bushell et al., 2022; Anstey et al., 2022), while climatological polar vortices in NH winter can be reproduced with observed strength. These results are consistent with previous QBO in multi-model ensemble studies that argued that unrealistically weak low-level QBO amplitude can weaken the QBO teleconnections to the polar vortex (Richter et al., 2022; Anstey et al., 2022c). In short, for any of the three experiments the models more often than not show a stronger polar vortex during NH winter when the 50-hPa QBO wind is westerly, and a weaker vortex when it is easterly, consistent with but weaker than the observed response.

Figure 3: (a) Monthly differences (QBO-W minus QBO-E) of zonal-mean zonal wind at 10 hPa averaged over 55°-65° N as a measure of the stratospheric polar vortex strength for the CTL experiment. QBO phases are classified same as Fig. 2. Solid dots show significant differences between QBO-W and QBO-E phases at the 90 % confidence level using a Monte Carlo test. Numbers in the legend are the cases included in each QBO phase. While for the experiments, ENSO is neutral, all years in ERA5 are included in the analysis (1959–2022). MMM means a multi-model mean. (b) Same as (a) but for the EN experiment including El Niño winters for ERA5. (c) Same as (a) but for the LN experiment including La Niña winters for ERA5. Numbers of (QBO-W, QBO-E) categories for ERA5 are (12, 11) in ENSO-Neutral, (7, 4) in El Niño, and (9, 4) in La Niña winters.

Intraseasonal Holton-Tan effects are investigated in Fig. 3, which shows the composite difference (QBO-W minus QBO-E) of monthly zonal-mean zonal wind at 10 hPa, 55°-65° N in CTL, EN, and LN experiments, together with ERA5. ERA5

presents a maximum Holton-Tan effect in January with a peak of 13 m s<sup>-1</sup> for the mean (dashed black line in the top panel), but this difference is lower in February during ENSO-neutral years (solid black line in Fig. 3a). Seasonal evolution of Holton-Tan effect is different between El Niño and La Niña winters; it seems stronger in early and late winters for the El Niño winters (middle panel) and in mid-winter for the La Niña winters (bottom panel), although it should be cautioned that the sample sizes (number of W/E winters) are small for both El Niño and La Niña groups. Some models show a similar seasonal cycle as ERA5 for their CTL runs (significant for MIROC-ESM and ECHAM5sh). Also, GISS in all months as well as LMDz and MIROC-AGCM in a few months exhibit an opposite sense to the observed Holton-Tan relationship for CTL. In EN, GISS, WACCM, EMAC, and ECHAM5sh capture the early-winter response in December although it is not statistically significant. The Holton-Tan relationship in El Niño years could depend on SSW occurrence because of the nonlinear joint effects of QBO and ENSO on the polar vortex as already explained in the Introduction. In LN, MRI-ESM2.0 and GISS capture the observed late-winter response relatively well, and other models do not show any response or even an opposite response.

### 375 3.2 SSW statistics






In this subsection, we examine SSW statistics modulated by ENSO and the QBO in the northern polar region. Previous observational studies indicated that the ratio of SSW frequency between La Niña and ENSO-neutral winters is dependent on details of the SSW definition (Butler and Polvani, 2011; Garfinkel et al., 2012a; Song and Son, 2018), and SSW statistics have been shown to depend on model biases (Tyrrell et al., 2022). Figure 4 shows frequencies of major and minor SSWs and final warming dates in NH for ERA5 and QBOi models. The approach to identify major, minor, and final warming dates is similar to what was proposed by previous studies (Charlton and Polvani, 2007; Butler et al., 2015). Major SSWs are identified when zonal-mean westerlies in winter are changed to easterlies at 60° N and 10 hPa. For minor SSWs, the zonal wind does not reverse but there is a change in sign of the meridional gradient of the zonal-mean temperature. Final warming date refers to the seasonal transition from westerly to easterly, with winds remaining easterly for the next months.

We consider first the influence of ENSO on SSW frequency. In ERA5 (the leftmost triplet of Fig. 4a panel), the frequency of major SSWs is high in ERA5 during both El Niño and La Niña years, compared to ENSO-neutral. Thus, we expect that major SSW frequencies in the QBOi models would be similar to the observations and have fewer events in CTL and more events in EN and LN experiments. LMDz and GISS reproduce the nonlinear observed ENSO response to some extent (Fig. 4a). However, most models show more SSWs during EN and they do not capture the LN response (e.g., EC-EARTH, MIROC-AGCM, MRI-ESM2.0). ECHAM5sh has similar frequencies in CTL and LN and more events in EN. GISS shows large spreads in CTL and EN, suggesting that the response is not statistically robust. In Fig. 4b, frequencies of minor SSWs are similar in both ENSO-neutral and El Niño years and there are fewer events in La Niña years in ERA5. There is a large spread in minor SSW frequencies between the models. EC-EARTH and ECHAM5sh show high frequencies of minor SSWs in EN whereas LMDz and MRI-ESM2.0 show low frequencies of minor SSWs. MIROC-AGCM produces fewer SSWs in the CTL, EN, and

LN experiments, and MIROC-ESM shows relatively higher frequencies for both major and minor SSWs in EN and LN compared to the other MIROC model. The GISS ensemble shows large spread in all three experiments, suggesting an important role for internal variability.




Figure 4: SSW statistics in NH in CTL, EN, and LN experiments for QBOi models including ENSO neutral, El Niño, and La Niña years for ERA5, based on their daily data. The order of triplets from left to right are La Nina (LN, purple), ENSO neutral winter experiment (CTL, grey), and El Nino (EN, brown). Frequency (number of events per decade) of (a) major SSWs (reversal of zonal-mean zonal wind at 10 hPa and 60° N; U60) and of (b) minor SSWs (reversal of 90°-60° N temperature gradient at 10 hPa without U60 reversal), occurring across full seasons. Different marker signs are used to indicate ensemble members, and uncertainties are estimated at the 5-95% level based on bootstrapping of 10 years of winter months. (c) Boxplots of final SSW date (day of year), considering full seasons, i.e., from westerlies onset to their turn to easterly for ERA5 and QBOi models based on their daily data.

The final warming date is when the transition from winter westerlies to summer easterlies occurs in the polar stratosphere (Butler et al., 2015). If the stratosphere is warmer in the polar regions, the transition of zonal wind to easterlies occurs earlier, and if it is colder the transition is delayed. Hence, we assume that in El Niño (La Niña) years when the polar stratosphere would be warmer (colder) as described in the Introduction, the final warming date might happen earlier (later). Consistent with this expectation, in ERA5 during La Niña (the leftmost triplet of Fig. 4c), the final warming date is more delayed than that in ENSO-neutral and El Niño years. GISS and MRI-ESM2.0 exhibit later final warming dates in LN than in EN, which is similar to the observed response (Fig. 4c). On the other hand, EC-EARTH, ECHAM5sh, LMDZ, MIROC-AGCM and MIROC-ESM do not show earlier final warming dates in EN, which is the opposite to the observed response. These results imply that the QBOi models have significant biases in reproducing observed SSWs statistics. Large inter-model variability is also diagnosed

by means of the Northern Annular Mode (NAM) index (Eyring et al, 2020) compositing at 500 hPa, as shown in Fig. S2, where the geopotential heights during LN tend to be lower and there are changes in the intensity of the extratropical signature between LN and EN. Inter-model variability in the large-scale response to ENSO may also explain the spread in the occurrence of SSWs due to differences in the simulated tropospheric forcing.

Figure 5: Scatter plots between mean vortex strength (60° N, 10 hPa) and major SSW frequency during DJF for different QBO and ENSO conditions. Major SSWs are identified as a reversal of daily zonal-mean zonal wind at 60°N and 10 hPa. QBO phases are classified using DJF-mean zonal-mean zonal wind anomalies at 50 hPa averaged over  $5^{\circ}$  S- $5^{\circ}$  N using  $\geq 0$  m s<sup>-1</sup> for QBO-W (WLY in panel) and  $\leq 0$  m s<sup>-1</sup> for QBO-E (ELY). The anomalies are calculated for each ensemble member of each experiment for the simulation data; those ones are calculated using all years (1959-2021 seasons) for the ERA5 data. For ERA5, El Niño and La Niña winters are when all three DJF months have the El Niño and La Niña flag, respectively. Number of (QBO-W, QBO-E) categories for ERA5 are (24, 15) in ENSO-Neutral, (5, 6) in El Niño, and (9, 4) in La Niña winters. For each condition, each model, the data are randomly resampled 100 times with replacement, and then 95% ranges are obtained and plotted.




Next, we investigate the influence of the QBO on major SSW frequencies modulated by ENSO in the NH winter. Figure 5 shows scatter plots between the climatological zonal-mean zonal wind at 60° N and 10 hPa and mean frequency of major SSWs in DJF during QBO-W and QBO-E years for three ENSO conditions. In ERA5, major SSW frequencies under QBO

and ENSO conditions are likely to be distinguishable. Averaged over all QBO conditions, the NH polar vortex is stronger in La Niña than El Niño winters, while SSW frequencies are slightly higher in La Niña than El Niño winters, and both are higher than ENSO-neutral winters. Major SSW frequencies in La Niña winters are significantly higher under QBO-E and lower under QBO-W, whereas those in El Niño winters are indistinguishable between QBO-W and QBO-E. Most QBOi models are characterized by linear distributions between SSW frequencies and the polar vortex strength. The EN experiment displays higher frequencies of SSWs than the LN experiment and SSW frequencies between QBO-W and QBO-E are indistinguishable. This shows that polar vortex responses to ENSO conditions in the QBOi models are stronger than responses to the QBOs in these models. Some models (EMAC, MIROC-AGCM, and MIROC-ESM) have very weak responses to both ENSO and QBO.

#### 440 4 The subtropical jet route of QBO teleconnections






This section examines the subtropical jet route of QBO teleconnection in the QBOi ENSO experiments. Only the late winter period of February to March, when the subtropical route is strongest in the observations (Garfinkel and Hartmann, 2011a; Park et al., 2022), is considered. Since the subtropical jet change in response to the QBO is pronounced for the APJ, analyses are performed for the zonal wind averaged over the Pacific sector (150° E–150° W). The sensitivity of the QBO-APJ connection to the ENSO phase is also examined.

The OBO-W minus OBO-E (W-E) composite differences are shown in Fig. 6 for the ENSO-neutral, El Niño, and La Niña winters, for both ERA5 and QBOi ENSO experiments. During the ENSO-neutral winter, the QBO W-E anomaly exhibits a distinct horseshoe-shaped pattern extending from the tropical lower stratosphere to the subtropical lower troposphere (top-left panel in Fig. 6). It is accompanied by a quasi-barotropic easterly anomaly in the extratropics. More importantly, the zonal wind anomalies switch sign across the climatological APJ (contour). This indicates that the APJ shifts equatorward during the OBO-W winter compared to the OBO-E winter. Most models underestimate or fail to reproduce the observed OBO-APJ connection. The dipolar wind anomalies are much weaker than those in observations in five models (i.e., EC-EARTH, ECHAM5sh, GISS, LMDz, and MIROC-ESM). Although one lobe of the dipolar wind anomalies is significant in ECHAM5sh and GISS, other models (i.e., EC-EARTH, LMDz, and MIROC-ESM) have statistically insignificant dipolar wind anomalies. MIROC-AGCM and MRI-ESM2.0 exhibit the opposite sign. Such large inter-model spread is consistent with a previous study (Anstey et al., 2022c). The QBO-APJ connection differs between El Niño and La Niña (top-middle and top-right panels in Fig. 6; Garfinkel and Hartmann, 2010). As the APJ strengthens over the Pacific sector (150° E-150° W) in response to El Niño (compare contours; Rasmusson and Wallace, 1983; Mo et al., 1998; Lu et al. 2008), the QBO subtropical wind anomalies become stronger near the APJ center during El Niño (top-middle panel; Ma et al., 2023). In contrast, the W-E anomalies switch sign across the climatological APJ during La Niña (top-right panel) as the APJ becomes slightly weaker (compare line contours in the top-left and top-right panels). The APJ's response to ENSO is consistently reproduced across models, whereas the ENSO modulation of the QBO-APJ connection shows large inter-model spread. While all models capture a stronger APJ during EN

than LN (compare line contours in the middle and right columns), they exhibit significant biases in reproducing the ENSO modulation of the QBO-APJ connection (filled contour).




Figure 6: QBO-W minus QBO-E composite differences of zonal wind averaged over the Pacific sector (150° E–150° W) for the ENSO-neutral (top-left), El Niño (top-middle), and La Niña (top-right) winters and those for the CTL, EN, and LN experiments (left to right columns). Values that are statistically significant at the 95% confidence level are cross-hatched. Contour denotes the climatological jet with zonal wind speed equal to or greater than 30 m s<sup>-1</sup>. QBO-W and QBO-E phases are defined as deseasonalized February-March zonal-mean zonal wind over 5°S–5°N at 70 hPa being > 0.5 σ and 

Figure 7: QBO-W minus QBO-E composite difference of the Asia-Pacific Jet (APJ) shift index. The APJ-shift index is defined as the difference of the 250-hPa zonal wind anomalies averaged over the Pacific sector (150° E–150° W) between the northern flank (40°–50° N) and the southern flank (20° N–30° N) of the climatological APJ core. The negative value indicates that the APJ moves toward the equator during the QBO-W phase. The composite differences are shown for the La Niña or LN experiment (blue), ENSO-neutral or CTL (black), and El Niño or EN experiment (red). The values are considered significant if the 5-95 % range of the bootstrap distribution (vertical dashed lines) does not encompass zero.

Given that the subtropical jet route of the QBO teleconnection can be influenced by the QBO amplitude and/or the climatological position of the subtropical jet (Garfinkel and Hartmann, 2011a), we examined whether model performance in simulating these two factors is related to the ability of model to capture the QBO-induced shift of the APJ (Fig. S3). Here, the QBO amplitude is defined as the root mean square of the deseasonalized zonal wind time series at 70 hPa, multiplied by  $\sqrt{2}$ , following Dunkerton and Delisi (1985) and Bushell et al. (2022). The climatological position of the APJ is defined as the latitude of the maximum zonal-mean wind averaged over the APJ sector (150° E–150° W). Consistent with previous studies (Bushell et al., 2022; Anstey et al., 2022c), most QBOi models underestimate the QBO amplitude. Only two models show a comparable QBO amplitude to the reanalysis. However, model biases in QBO amplitude do not affect those in the QBO-APJ connection (Fig. S3a). Models with larger QBO amplitudes do not necessarily exhibit stronger jet responses, nor do models

with smaller amplitudes consistently show weaker responses. A similar result is also found in the APJ position (Fig. S3b).

These results suggest that neither the QBO amplitude nor the APJ position explains the inter-model spread in the QBO-APJ connection. Other factors, such as transient and stationary eddies, may determine the QBO-APJ connection in the model. This possibility needs to be explored in a future study.

# 5 QBO teleconnections: the tropical route

This section investigates the tropical route of the QBO teleconnection modulated by ENSO, focusing on tropical precipitation and the Walker circulation.

# 5.1 Tropical precipitation





Figure 8 shows the DJF seasonal-mean precipitation differences between QBO-W and QBO-E in EN and LN, together with anomalies for El Niño and La Niña winters for GPCP. In the observations, the QBO signals are largest and statistically significant in the tropical Pacific and Indian oceans, and are in good agreement with previous analyses (Liess and Geller, 2012; García-Franco et al., 2022). The positive equatorial Pacific signal in the GPCP dataset, which resembles an El Niño anomaly (Dommenget et al., 2013; Capotondi et al., 2015), is particularly strong and statistically significant in DJF. This signal is associated with the three strongest El Niño events (1982–1983, 1997–1998, 2015–2016) coinciding with the westerly QBO phase (Fig. S4 and García-Franco et al., 2023).

Although most models do not show such El-Niño-like precipitation anomaly patterns in either experiment, several models exhibit significant precipitation QBO-related signals. For example, GISS, ECHAM5sh and MRI-ESM2.0 show significant QBO responses in the EN experiment, which are comparable in magnitude to the signal diagnosed in GPCP when considering all months (Fig. S5a) but weaker than the corresponding observed signals in El Niño and La Niña conditions. However, the response in other models is generally weaker, and the spatial distribution of the anomalies is not consistent between models. In the LN experiments, the models similarly do not show a clear precipitation signal in the Pacific, but EC-EARTH, ECHAM5sh, WACCM and MIROC-ESM show several precipitation signals over the Indian Ocean and Australia. A multimodel mean response is shown in Fig. S5, which illustrates the lack of model agreement characterized by a virtually zero QBO signal in a multi-model mean sense across the tropics. Thus, this result suggests that there is a lack of model agreement in the spatial distribution and sign of the tropical precipitation response to the QBO phase. These results are also supported by Fig. S6, which shows DJF seasonal outgoing longwave radiation (OLR) differences between QBO-W and QBO-E in EN and LN together with ERA5. None of the models show an OLR signal comparable to observations, and models show a distinct QBO signal between EN and LN experiments. In other words, there is no consistent or robust precipitation response across models or experiments.

Figure 8: DJF seasonal mean precipitation differences (mm day $^{-1}$ ) (QBO-W minus QBO-E) for (left) EN and (right) LN experiments for the QBOi models including El Niño and La Niña years in GPCP data. Hatching denotes statistical significance at the 95% confidence level according to a bootstrap test for the observations and a two-sided t-test for the models. The observed composite sizes in months are shown in parenthesis in the GPCP panels. QBO phases are classified using deseasonalized DJF mean zonal-mean zonal wind averaged over 5° S-5° N at 50 hPa using  $\geq$  2 m s $^{-1}$  for QBO-W and  $\leq$  -2 m s $^{-1}$  for QBO-E.

Figure 9: (a-b) Box plots of QBO-W minus QBO-E differences in DJF precipitation (mm day<sup>-1</sup>) in (a) the western equatorial Pacific (WEP) and (b) EN3.4 region (5° S-5° N, 170°-120° W). Error bars represent the 95 % confidence interval. Note that the y-axis is fixed to make the plot clearer, but an alternative version where the y axis limits are set based on the GPCP bar is provided in supplementary Fig. S8.

Previous studies have shown that the QBO signal in DJF is prominent in particular regions of the tropical Pacific: the western equatorial Pacific (WEP) region (5° S–5° N, 120°–170° E) and the Nino3.4 region (5° S–5° N, 170°–120° W) (EN3.4; Gray et al., 2018, Serva et al., 2022, García-Franco et al., 2022). To examine the extent to which precipitation in these regions is sensitive to the QBO phase, we evaluated the area-averaged precipitation in both regions as a function of QBO and ENSO phases (Fig. S7). The QBOi models show significant spread in their climatology of precipitation amounts but all the simulations seem to reproduce the observed ENSO signal, i.e., wetter conditions in the EN3.4 region and drier in the WEP in EN and the opposite in LN, regardless of the QBO phase.

Figure 9 shows the area-averaged precipitation differences (QBO-W minus QBO-E) per region for the in CTL (Neutral), El Niño and La Niña experiments/winters.) In observations, the precipitation signal associated with the QBO during El Niño is opposite in sign to that of La Niña. One must consider the very small sample size (roughly 3 to 5 winters in each composite) in these observations when interpreting these results. Regardless of the sign and magnitude of the observed response, the

models seem to show disagreement on the sign of the precipitation response, i.e., comparing models in the same experiment provides no consistent precipitation signal. For example, while the La Niña response is positive over the WEP in observations and most models agree, only 5 out of 7 models show a positive response. When looking at individual models, GISS and MIROC-ESM agree that the precipitation signal (QBO-W minus QBO-E) is positive in the WEP in all their three experiments but no model agrees on the sign of the precipitation response in all three experiments for the EN3.4 region. These results emphasize that the QBO signal on tropical precipitation may strongly depend on the state of ENSO as suggested by the observations (García-Franco et al., 2023). Overall, some models show a robust and significant precipitation response to the QBO but this response is distinct from observations and varies in sign and magnitude amongst experiments and models.

Figure 10: Scatter plot of DJF air-temperature differences at 100 hPa (QBO-W minus QBO-E in K) versus precipitation differences (QBO-W minus QBO-E in mm day⁻¹). Both variables were averaged in the western equatorial Pacific region. The correlation of the best-fit line for all the data, including observations, is −0.48, which is significant to the 95% confidence level according to a t-test. (The correlation without observation is −0.25.) The coefficient changes when only El Niño conditions are considered (r = −0.82) as well as for La Niña experiments (r = −0.2).

One reason for this inter-model and/or inter-experiment spread in the precipitation response could be the model spread in QBO-related temperature-associated anomalies at the equator resulting from the QBO mean meridional circulation and thermal wind balance. The QBO impact on the tropical tropopause layer (TTL) region is important for the QBO teleconnection in the tropical route (Haynes et al., 2021, Hitchman et al., 2021). Here, one common hypothesis is that if a cold QBO anomaly lies in the TTL, convective systems may grow more efficiently, penetrating to greater altitudes, locally amplifying the zonal mean QBO cold anomaly (Tegtmeier et al., 2021). Figure 10 shows a scatter plot of the QBO-W minus QBO-E difference of air temperature at 100 hPa versus the precipitation difference (QBO-W minus QBO-E) over WEP. One could reasonably question

whether models or experiments with a larger TTL temperature signal or static stability may also show a larger signal in precipitation. In this panel, the W–E TTL temperature signals diagnosed from ERA5 are larger than those of the models, and are strongest for El Niño. The precipitation signal diagnosed in GPCP is also largest in El Niño, possibly due to the coincidence of the largest El Niño events with the westerly QBO phase. We confirmed an impact of removing those strongest El Niño events (1982–1983, 1997–1998, 2015–2016) on the GPCP precipitation signal (Fig. S4). It is found that the impact is more dramatic over the all-year composites at the top, in which the Pacific signal disappears when not considering these cases (Fig. S4a,b). In the El Niño winters, it is only the eastern portion of the Pacific that changes significantly. There are some models that have a strong temperature signal, such as GISS and ECHAM5sh, which also have a strong negative precipitation signal in LN. However, most models have modest positive temperature differences without a clear precipitation signal. Overall, the QBOi models show unrealistically weak QBO wind amplitudes in the lower stratosphere (Bushell et al., 2022), and correspondingly have temperature anomalies that are too weak in the TTL (Serva et al., 2022), which could help explain the weak precipitation signals.

#### 5.2 Walker circulation

In this subsection, we examine whether a QBO impact on the Walker circulation can be detected across different ENSO phases. A recent study (Rodrigo et al., 2025) showed that in reanalyses the QBO signal in the divergent circulation is strongest over the Maritime Continent region in boreal summer (JJA), followed by autumn (SON), and weakest in winter. However, under El Niño and La Niña conditions this timing may slightly shift, potentially due to the ENSO influence on the QBO itself (Taguchi, 2010b; Kawatani et al., in revision). Additionally, model diversity and biases in the simulated QBO (Bushell et al., 2022) could lead to inter-model variations in the simulated QBO teleconnection. We begin the analysis by applying a common QBO definition and target season to all models, using the zonal-mean zonal wind at 70 hPa during JJA to define the QBO. With this approach, we identify a coherent signal, characterized by anomalous westerlies in the upper troposphere and anomalous easterlies in the lower troposphere over the Indian Ocean–Maritime Continent, in observations and some models in CTL, LN, and EN experiments (Figures S9, S10 and S11). To enhance this signal and capture the strongest response in each model, we allow slight adjustments to the QBO definition and target season when necessary. The Bonferroni correction (Holm, 1979; see Section 2) is applied to the two-sided *t*-test when a level or season other than 70 hPa during JJA is used to define the QBO phase.

Figure 11 shows the QBO zonal wind signal averaged over 10° S–10° N in the LN experiment, represented by the QBO-W minus QBO-E composite (shading), with the climatological winds superimposed (black contours). Focusing on ERA5 during La Niña (Fig. 11a), the August-September-October (ASO) mean state features upper-level easterlies over the Eastern Hemisphere and westerlies over the Western Hemisphere, with a weaker, opposite pattern in the lower troposphere. A distinct QBO signal is observed in the equatorial troposphere over the Indian Ocean–Maritime Continent. This signal is characterized by anomalous westerlies in the upper troposphere (red contours and shading) and anomalous easterlies in the lower troposphere

(blue contours and shading). Relative to the climatology, this signal represents a weakening of the mean zonal circulation over the Indian Ocean–Maritime Continent region. Similar QBO-related anomalies to those observed in ERA5 for La Niña, featuring upper-level westerlies and lower-level easterlies, are also found in most models for LN (Figs. 11b-i), although their precise location varies and the lower-level anomalies are generally weaker. Specifically, the strongest signals are identified in EC-EARTH, MRI-ESM2.0, LMDz and MIROC-AGCM during JJA; GISS during SON; and in WACCM during MJJ. In contrast, ECHAM5sh and MIROC-ESM exhibit no significant signal. The QBO-W minus QBO-E composite in CTL shows a similar signal to that in LN in most models during summer or autumn (Fig. S12). This modulation of tropical circulation by the QBO appears robust, despite differences in the timing and longitudinal location.

Figure 11: Climatology (black contours) and QBO Westerly (W) minus Easterly (E) differences (shading and colored contours) in equatorial zonal wind profiles, averaged over 10° S–10° N, from the LN experiment for the QBOi models. Black contours are drawn at 4 m s<sup>-1</sup> intervals, and the colored contour intervals match the shading scale in the color bar. The target season for each panel is indicated in the title, with the QBO definition provided in the legend. In ERA5, 15 La Niña events are identified using the NINO3 index during DJF, and they are classified into 10 QBO-W and 5 QBO-E categories by analyzing the zonal-mean zonal wind at 50 hPa in summer and autumn. Only statistically significant zonal wind differences at the 95% confidence level are shaded. For models with a QBO definition other than 70 hPa during JJA, the Bonferroni correction is applied (see Section 2). Note that the color bar for ERA5 differs because of the larger QBO amplitude.

Figure 12: Same as Figure 11, but for EN experiments and El Niño years in ERA5. In ERA5, 14 El Niño events are identified, and they are classified into 7 QBO-W and 7 QBO-E categories.

During El Niño in ERA5 (Fig. 12a), the QBO signal in the equatorial troposphere resembles that observed during La Niña, although it occurs during JJA and is weaker. It also shows anomalous westerlies in the upper troposphere over the Indian Ocean–Maritime Continent and anomalous easterlies in the lower troposphere. As for LN, this anomalous zonal circulation implies a weakening of the climatological pattern. Comparable anomalies, featuring upper-level westerlies and lower-level easterlies over the same region are also present in most models. The strongest signals occur in EC-EARTH during MJ; in MRI-ESM2.0, GISS, LMDz, MIROC-AGCM and MIROC-ESM during JJA; and in WACCM during JAS. In contrast, ECHAM5sh displays a weak response that differs from the other models.

Figure 13 shows a summary diagram of when and where the statistically significant composite differences in equatorial zonal wind (10° S–10° N) occur across all three experiments, illustrating QBO-W minus QBO-E differences at three representative vertical levels (700, 100 and 70 hPa) over four standard seasons. These statistically significant signals are identified by examining the influence of QBO on zonal winds within the longitudinal band from 60° E and 120° E. An example from the EC-EARTH CTL experiment is provided in Fig. S13. The QBO phase is consistently defined in the specific season indicated in the legend (i.e., it does not vary seasonally). In some models, the strongest signals occur during transitional periods between standard seasons, so the corresponding symbols are placed accordingly. Across all three experiments, nearly all models, along with ERA5, exhibit a tropospheric signal characterized by upper-level (100 hPa) westerlies and lower-level (700 hPa) easterlies during varying seasons from May to November, suggesting a weakening of the climatological Walker circulation over the Indian Ocean–Maritime Continent. Exceptions include GISS in CTL, MIROC-ESM in CTL and LN, and ECHAM5sh in LN and EN (see Figs. 11, 12 and S12). Overall, this figure illustrates that the QBO, when defined around

summer and autumn, modulates the zonal circulation in the equatorial troposphere over the Indo-Pacific region. ERA5 shows a consistent signal during both La Niña and El Niño years, which is reproduced in some models with slight variations in season, longitude, or the level used to define the QBO, but missing in others. Again, we note that the QBOiENSO experiments are idealized and ERA5 should not be considered a true benchmark.

Figure 13: (a) Occurrence of a statistically significant zonal wind signal by models, season and altitude over the equatorial (10° S–10° N) 60° E–120° E band for the (a) CTL, (b) LN, and (c) EN experiments. The QBO-W minus QBO-E zonal wind signals are evaluated at three vertical levels and across four standard seasons. Symbols are placed between standard seasons when the strongest signal occurs in an intermediate period. Filled symbols represent westerly anomalies, while open symbols indicate easterly anomalies. The QBO definition for each model and experiment is provided in the legend.

760

## 6 Summary and Discussion

780

790

795

In this paper, we have examined QBO and ENSO teleconnections in the Arctic stratosphere, the subtropical Pacific jet, and the tropical troposphere. A multi-model ensemble of QBO-resolving models that performed the QBOiENSO experiments has been used to examine the robustness of these teleconnections. Difficulties can arise in distinguishing QBO and ENSO influences on the extratropics and tropical troposphere due to the observed aliasing between the QBO and ENSO. Here we have attempted to separate these competing influences by conducting model integrations with annually-repeating, prescribed SSTs that are characteristic of either strong El Niño or La Niña conditions, thereby simplifying the ENSO forcing in comparison to the diversity of observed ENSO phases. We have reexamined QBO teleconnections to the extratropics and tropics that were explored in previous QBOi studies (Anstey et al., 2022c; Serva et al., 2022) but now addressing combined QBO-ENSO influences using this new QBOi dataset of idealized ENSO experiments.

The observed strength of correlation coefficients between 50-hPa equatorial zonal wind and the polar vortex strength at stratospheric altitudes in DJF shows large uncertainty (Fig. 1a) but the confidence intervals clearly exclude zero at most altitudes during La Niña and ENSO-neutral winters, while El Niño response is statistically significant over a smaller altitude range. The models show a smaller uncertainty due to their larger sample sizes (Fig. 1). Some models have weaker correlations for a particular ENSO experiment, similar to the observations. The Holton-Tan relationship in ERA5 represents the polar vortex being significantly stronger (weaker) under QBO-W (QBO-E) for all the ENSO phases, with the strongest response occurring in the La Niña phase. Nearly half of the models exhibit stronger polar vortex during NH winter under QBO-W for each experiment, consistent with, but much weaker than the observed response (Fig. 2). Seasonal evolution in ERA5 indicates a stronger signal in early (late) winter for the El Niño (La Niña) winters. In LN, two out of nine models capture the observed late-winter response relatively well, and other models do not show any response or even the opposite direction (Fig. 3).

Major SSWs occur frequently during both El Niño and La Niña winters, compared to ENSO-neutral, in ERA5. Most models show more events during EN but they do not catch the LN response, implying that the QBOi models have some trouble in reproducing observed SSWs statistics (Fig. 4). Major SSW frequencies in ERA5 show strong variation with QBO and ENSO phase. QBOi models are characterized by linear distributions between SSW frequencies and the polar vortex strength in NH winters (similar to ERA5) and overall the EN (LN) experiment displays high (relatively low) SSW frequencies (Fig. 5). SSW frequencies between QBO-W and QBO-E are indistinguishable in the models, indicating that polar vortex responses to the idealized ENSO forcing in the QBOi models are strong, while vortex responses to equatorial QBOs are fairly weak.

The APJ changes in response to the QBO are investigated (Figs. 6 and 7), focusing on the late winter when the subtropical route is strongest in the observations. In observations, the QBO westerly anomaly exhibits a distinct horseshoe-shaped pattern extending from the tropical lower stratosphere to the subtropical lower troposphere, indicating that the APJ shifts equatorward during the QBO-W winter compared to the QBO-E winter. However, most models underestimate or fail to reproduce the observed QBO-APJ connection. The observed QBO-APJ connection differs between El Niño and La Niña. In observations, as the APJ strengthens over the Pacific sector in response to El Niño, the QBO subtropical wind anomalies become stronger near

the APJ center during El Nino while they do not change much during La Niña as the APJ becomes slightly weaker. All models capture a stronger APJ in EN than in LN.

The positive equatorial Pacific signal in the GPCP dataset, which resembles an El Niño anomaly for W-E differences, is particularly strong and statistically significant in DJF, as shown by previous studies that highlight the issue of strong ENSO events coinciding with the westerly phase (García-Franco et al., 2023). Although most of the models do not show such El-Niño-like precipitation anomaly patterns in either EN or LN experiments, some models (EC-EARTH, ECHAM5sh, WACCM and MIROC-ESM) show significant precipitation signals over the Indian Ocean and Australia (Fig. 8). The precipitation response to the QBO in these experiments depends on both the model, region and ENSO phase, as there is no consistent response between the experiments for each model (Fig. 9). For example, the simulated and observed QBO signals in the Niño 3.4 region disagree on the magnitude and sign. To explore the causes of model versus observation differences, the strength of the QBO impact on the TTL region was analyzed as it is considered to be important for the QBO teleconnection in the tropical route (Fig. 10). In particular, we verified whether the strength of the temperature anomaly could explain inter-model or inter-experiment differences in the precipitation signals. Overall, the QBOi models have too-weak wind amplitudes and too-weak temperature anomalies in the lower stratosphere, which could help explain the weak precipitation signals.

Several potential biases likely influence the tropical route of QBO teleconnections. Most proposed mechanisms linking the QBO to the tropical surface rely on interactions between the lowermost stratosphere and the uppermost troposphere. A key bias common to many models, including those used in this study, is a weak QBO amplitude in the lower stratosphere, which limits the effectiveness of stratosphere–troposphere coupling processes (Oueslati et al., 2013; Richter et al., 2020; García-Franco et al., 2022, 2023). Additionally, models exhibit persistent tropospheric biases related to tropical convection and precipitation, including biases in the strength and position of the ITCZ, tropical wave activity and unrealistic distributions of rainfall intensity (Oueslati et al., 2013, Norris et al., 2021). These biases typically stem from model parameterizations, notably in convection and cloud microphysics schemes (Hagos et al., 2021, Norris et al., 2021, Zhou et al., 2022). The combination of these stratospheric and tropospheric biases likely weakens the QBO signal reaching the tropical troposphere and contributes to inter-model differences in the magnitude, timing and spatial manifestation of the teleconnection.

The QBO teleconnection to the Walker circulation in reanalyses is strongest over the Indian Ocean–Maritime Continent region in boreal summer, followed by autumn, and weakest in winter (Rodrigo et al., 2025). Under ENSO conditions, this timing may slightly shift, potentially due to the influence of ENSO on the QBO itself. Furthermore, model diversity and biases, as described above, may cause the simulated QBO teleconnection to vary. Here, we identified the strongest signal for each model, defining the QBO across different seasons (JJA or SON) and vertical levels (85 or 70 hPa). A distinct QBO signal, characterized by upper-level westerly and lower-level easterly anomalies, is observed in the equatorial troposphere in ERA5 during both El Niño and La Niña years. Most models reproduce a similar pattern across all three experiments, although the lower-level anomalies are generally weaker. This modulation of the tropical circulation by the QBO appears spatially robust, but its timing varies.

We now consider three issues about modelling QBO-ENSO complexity raised by these results: forced SSTs, the seasonality and variation of the Walker circulation, and biases in the OBO and other diagnostics. AMIP-type experiments, where idealized SST patterns and fixed external forcings are used, have been used here to examine QBO-ENSO teleconnections although it is noted that we do not have an observational verification for these experiments. However, the responses of the climate system to ENSO forcing tend to be nonlinear with respect to ENSO intensity and asymmetric with respect to the polarity of ENSO (Domeisen et al., 2019; Rao and Ren, 2016b, c). This means that it is difficult to isolate physical meaningful mechanisms from such a nonlinear system and gain scientific insights into OBO-ENSO teleconnections. Conducting idealized experiments that take into account the ENSO-QBO diversity could help us to further elucidate scientifically meaningful mechanisms in such a complex system. It is noted that the experimental design of QBOiENSO (Kawatani et al. in revision) is annually locked with monthly-mean anomalies from the climatology. For example, the precipitation responses to the QBO for the AMIP-type experiments with interannually varying SSTs (Serva et al., 2022; García-Franco et al., 2022) is different from those for the OBOi ENSO experiments with perpetual SSTs. The precipitation response to the OBO in the equatorial Pacific signal in the GPCP dataset shows a statistically significant, El-Niño-like anomaly pattern. Most of the models do not show such El-Niñolike precipitation anomaly patterns in the CTL, EN or LN experiments, while such patterns were seen in some of the QBOi models in the QBOi Experiment 1 (Serva et al., 2022). The lack of a robust and coherent QBO-related precipitation signal across experiments and models highlights significant spread in how convection and circulation respond to a QBO forcing. This raises the possibility that the OBO's downward impact on tropical precipitation is too sensitive to model physics, or is perhaps muted by the lack of SST feedbacks (García-Franco et al., 2023, Randall et al., 2024) to be clearly detected.








One of the most important points from this study is that the Walker circulation would potentially play an important role in tropical teleconnections as well as extratropical teleconnections. We are interested in two distinct and documented El Niño patterns, Eastern Pacific (EP) versus Central Pacific (CP, or Modoki) El Niños, which make a large difference in the Hadley and Walker circulations and also have markedly different impacts on remote regions. One may doubt that weaker ENSO events or different ENSO flavors than those used in this study would yield further insights due to such ENSO events being associated with less dramatic changes in the location of tropical convection. However, the tropical SSTs in the Central Pacific substantially influences the QBO on decadal timescales (Shibata and Naoe, 2022). Thus, such idealized experiments forced with ENSO SST patterns would be beneficial for us to better achieve the changing impact of ENSO events on the QBO teleconnections. We are also interested in tropical convection being inherently coupled with the ocean. Long-term simulations from coupled global circulation models (CGCMs) would be a convenient tool for testing responses of QBO-ENSO teleconnections associated with internal variability of the ocean-atmosphere coupled system (García-Franco et al., 2023; Randall et al., 2023).

We have to underline the importance of seasonality for the combined effect of QBO-ENSO on the tropical troposphere. Our results suggest that QBO teleconnections with the Walker circulation exhibit seasonal variability and a distinct zonally asymmetric pattern. These findings emphasize the need for further investigation to elucidate the drivers of the seasonal dependence, the nature of the asymmetry and the underlying mechanisms governing these interactions.

In the extratropical stratosphere, the previous studies of QBOi models suggested that the systematic weakness of the QBO-polar vortex coupling in the models might arise from systematically weak QBO amplitudes at lower levels in the equatorial stratosphere, polar vortex biases in winter, inadequate representation of stratospheric-troposphere coupling, etc. (Bushell et al., 2022; Richter et al., 2022; Anstey et al., 2022c). In our QBOiENSO experiments, such systematic model biases were also found because most of the modes were the same as the previous studies. In the tropics, our results suggested that the systematic bias of the QBO impact on the tropical troposphere might arise from the systematically weak QBO amplitudes at lower levels, precipitation bias, and inadequate representation of the Walker circulation. Thus, the combination of several biases could be the reason why we have not seen a consistent signal of QBO teleconnections across the models and experiments. Therefore, it is plausible that consistency with observations will not improve without correcting such model biases. Currently, a project of QBOi Phase 2 is in progress to assess the impact of QBO biases by using zonal mean nudged toward observations in the QBO region. Bias-corrected QBO amplitude, achieved through nudging methods, may provide insights for improving the representation of QBO teleconnections to the extratropics and the tropical troposphere.

Data availability. The QBOi data archive was hosted by the Centre for Environmental Data Analysis (CEDA), UK, and processing was performed on the JASMIN infrastructure. The ERA5 reanalysis data can be obtained from the ECMWF website (https://www.ecmwf.int/en/forecasts/datasets/browse-reanalysis-datasets). The El Niño Monitoring Index (NINO.3) **ENSO** was derived from an monitoring site of the Japan Meteorological Agency (https://ds.data.jma.go.jp/tcc/tcc/products/elnino/index.html). The **GPCP** v2.3 was downloaded from https://doi.org/10.7289/V56971M6 (Adler et al., 2016).

Author contribution. YK, KH, JA, JHR designed the QBOi ENSO-QBO experiments. FS, HN, KY, TK, SW, YK, JGS, FMP, PAB, FL, CO, JHR, CCJ created model-experimented data and uploaded them to the CEDA. HN, JLGF, CHP, MR, FMP, FS and MT conducted the analyses, prepared the figures, and contributed to the manuscript. JA, JGS, SWS, NB, and SO contributed in the interpretation of results. The first draft of the manuscript was prepared by HN, who involved all authors for the final version.

Competing interests. The authors declare that they have no conflict of interest.







Acknowledgements. All authors acknowledge the CEDA, which kindly hosted the QBOi data archive. HN was supported by MEXT, JSPS KAKENHI (grant numbers: JP22H04493, JP24K07140); HN and KY were supported by JSPS KAKENHI JP24K00710. YK was supported by JSPS KAKENHI (JP22K18743) and the Environment Research and Technology

Development Fund (JPMEERF20242001) of the Environmental Restoration and Conservation Agency provided by Ministry of the Environment of Japan. YK and SW were supported by JSPS KAKENHI (JP22H01303 and JP23K22574). SW was supported by MEXT-Program for the advanced studies of climate change projection (SENTAN) Grant Number JPMXD0722681344. The numerical simulations of MIROC models were performed using the Earth Simulator. CP and SS were funded by the Korea Meteorological Administration Research and Development Program under Grant (RS-2025-02307979). MR was supported by the 'Ayudas para la Formación de Profesorado Universitario' programme (FPU20/03517), FMP was supported by the EU/HORIZON-funded MSCA-IF-GF SD4SP project (GA 101065820), and JGS acknowledges funding from the Spanish DYNCAST project (CNS2022-135312). The ECHAM5sh simulations were performed thanks to an ECMWF Special Project awarded to FS. NB was funded by the Met Office Climate Science for Service Partnership (CSSP) China project under the International Science Partnerships Fund (ISPF).

Review statement. This paper was edited by xxx and reviewed by xxx referees.

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
