# Peer review of "QBOi El Niño Southern Oscillation experiments: Teleconnections of the QBO"

_EGUsphere, 2025_

## Editor Comment (EC1)

Dear Dr. Naoe and colleagues:

Both Anonymous referees have posted their comments on your manuscript (WCD 2024-1148). As per WCD policy, you are now to post a response on how you will address the referee's comments – after which I will make a decision on the manuscript.

Both reviewers have made excellent comments on the manuscript and call for revisions (one major, one minor). To provide guidance in revising the manuscript so that it is acceptable for publication in WCD, below I itemize the issues that I expect will be addressed in a revised manuscript. I will also post these on the WCD page for the manuscript.

Both anonymous reviewers feel this is a worthwhile manuscript for publication in WCD, and I agree. The opening paragraph by Reviewer #1 has a very succinct summary of the paper, followed by 8 bulleted points that contain either comments or suggestions. I strongly recommend you address all the comments, and adopt all the suggestions. In particular, the reviewer notes that the text is not sufficiently critical of the model results concerning the impact of the QBO on the polar vortex (Figs. 1 and 2), stating: "only ECCAM5, WACCM and MRI are reasonably correct for neutral ENSO, but none get El Nino right. Maybe ECCAM5 and MRI get LaNina right (relative to ERA5)." I agree: only two of the models get within ½ the amplitude of the observed for the ENSO neutral case (MRI and WACCM), and only the MRI model also shows a stronger impact on the QBO on the vortex under La Nina conditions than under El Nino conditions (but even that model has the wrong sign response for El Nino conditions). The reviewer also asks for more clarification on the text on lines 221-226, and clarification on the statistical significance when multiple indices are used in the identification of the QBO. Reviewer #1's suggestion "to include more in-text references to figure panels being discussed" would really help the reader.

Reviewer #2 also has excellent major comments and I strongly recommend you address them in your revised manuscript. In particular, Reviewer #2 asks for more discussion and analysis of why almost all the models do not reproduce three of the four teleconnections examined, and I agree. In some cases, further analysis may be required to support these discussions (e.g., is there a relationship between the model biases in the strength of the simulated QBO (in either neutral, El Niño and La Niña conditions) and the strength of the polar vortex response? Is there a relationship between the model biases in the strength of the polar vortex and the polar vortex response? Is there a relationship between biases in the extratropical stratospheric winds and the weakness in the impact of the QBO phase on the polar vortex?).

Reviewer #2 also notes that previous work suggested that a measure of the efficacy of a model to reproduce QBO's impact on the polar vortex (the Holton-Tan effect) seen in observations is sensitive to the level that is used as an index of the QBO, and that model differences in the QBO justify the use of model-specific indices. Please address this point in your revised manuscript. Also, if you did choose levels to define the QBO that were model-specific, would the QBOs simulated by the models still be only half as strong as that observed (as documented in Fig. 3)? Would that still be the leading candidate for the weak relationships between the QBO phase, ENSO phase and polar vortex?

Finally, Reviewer #2 asked why a different analysis procedure was used to examine the relationship between the QBO phase and the Walker circulation than that used to examine the other three teleconnections and whether the teleconnections were stronger for the Walker circulation simply because optimal pressure levels and seasons were chosen. I am not to bothered by this because, to be frank, the evidence presented in this section is pretty damning. Contrary to the description in the text, the observed relationship between the Walker circulation and the phase of the QBO shown is not well reproduced by most of the models for either La Nina or El Niño conditions. For La Nina conditions (Fig. 11), the anomalies in the zonal winds over the Pacific show a slightly westward shifted Walker circulation, whereas the models b,d,g,h and i shows a weakened Walker circulation (in phase anomalies of the opposite sign as the climatology aloft) and model e shows only easterly anomalies everywhere. The agreement during El Nino conditions is even worse (Fig. 12). [By the way, please note the contour interval for the anomalies in these figures. They seem to be much coarser than the discretized colorbars.] Stepping back a bit, I wonder whether the relationship between the QBO phase and the Walker circulation is poor because the band to define (5S-5N) the Walker circulation may be too narrow; 10S to 10N would better capture the zonal wind anomalies associated with the Walker circulation. Based on Fig. 17.17 of Wallace et al (2023), I expect this isn't the answer -- but it might be worth checking.

Minor comments:

Does the GPCP bar in panel 9b stop the top of the plot, or does it run off scale? Why isn't there an error bar on this bar?

In Fig. 10, is the temperature also the zonal mean over the western Equatorial Pacific, or is it a zonal mean?

Lines 777-790: These statements are inconsistent with the published papers, dating back as far as Hoerling et al. (1987). Atmosphere general circulation models DO robustly reproduce the nonlinearity in the atmospheric response to warm and cold ENSO phases, given El Niño and La Niña SST anomalies.  Also, given the very weak relationship between the QBO phase, the ENSO phase, and the tropical anomalies shown in this study, it is unlikely that weaker ENSO events or ENSO events with less dramatic changes in the location of tropical convection than used in this study would yield further insights.

References

Hoerling, M. P., Kumar, A., & Zhong, M. (1997). El Niño, La Niña, and the nonlinearity of their teleconnections. *Journal of Climate*, **10**(8), 1769–1786. **https://doi.org/10.1175/1520-0442(1997)010<1769:ENOLNA>2.0.CO;2**

Wallace et al, "The Atmospheric General Circulation", Cambridge University Press, 2023.

---

## Author Response (AR1)

**Response to Editor**

Title: QBOi El Niño Southern Oscillation experiments: Teleconnections of the QBO

Authors: Hiroaki Naoe, et al.

WCD manuscript on EGUsphere, MS No: egusphere-2025-1148

The authors would like to thank both Reviewers and the Editor for their time and effort in reviewing our manuscript entitled "QBOi El Niño Southern Oscillation experiments: Teleconnections of the QBO". Above all, the authors are deeply grateful for the many insights gained by reading the papers recommended by the reviewers. We incorporate their valuable comments and suggestions on how our proposed revised manuscript addresses their concerns. Our reviewer responses and revision are shown in blue text whereas reviewers' and Editor's comments are shown in black. Individual responses to the Editor are as follows.

Black: Editor's comments

Blue: Authors response to the Editor

**To Editor:**

EC1: 'Editor Comment on egusphere-2025-1148', David Battisti, 17 May 2025

Dear Dr. Naoe and colleagues:

Both Anonymous referees have posted their comments on your manuscript (WCD 2024-1148). As per WCD policy, you are now to post a response on how you will address the referee's comments — after which I will make a decision on the manuscript. Both reviewers have made excellent comments on the manuscript and call for revisions (one major, one minor). To provide guidance in revising the manuscript so that it is acceptable for publication in WCD, below I itemize the issues that I expect will be addressed in a revised manuscript. I will also post these on the WCD page for the manuscript.

Both anonymous reviewers feel this is a worthwhile manuscript for publication in WCD, and I agree.

The opening paragraph by Reviewer #1 has a very succinct summary of the paper, followed by 8 bulleted points that contain either comments or suggestions. I strongly recommend you address all the comments, and adopt all the suggestions. In particular, the reviewer notes that the text is not sufficiently critical of the model results concerning the impact of the QBO on the polar vortex (Figs. 1 and 2), stating: "only ECCAM5, WACCM and MRI are reasonably correct for neutral ENSO, but none get El Nino right. Maybe ECCAM5 and MRI get LaNina right (relative to ERA5)." I agree: only two of the models get within 1/2 the amplitude of the observed for the ENSO neutral case (MRI and WACCM), and only the MRI model also shows a stronger impact on the QBO on the vortex under La Nina conditions than under El Nino conditions (but even that model has the wrong sign response for El Nino conditions). The reviewer also asks for more clarification on the text on lines 221-226, and clarification on the statistical significance when multiple indices are used in the identification of the QBO. Reviewer #1's suggestion "to include more in-text references to figure panels being discussed" would really help the reader.

In particular, the reviewer notes that the text is not sufficiently critical of the model results concerning the impact of the QBO on the polar vortex (Figs. 1 and 2), stating: "only ECCAM5, WACCM and MRI are reasonably correct for neutral ENSO, but none get El Nino right. Maybe ECCAM5 and MRI get LaNina right (relative to ERA5)." I agree: only two of the models get within? the amplitude of the observed for the ENSO neutral case (MRI and WACCM), and only the MRI model also shows a stronger impact on the QBO on the vortex under La Nina conditions than under El Nino conditions (but even that model has the wrong sign response for El Nino conditions).

Thank you very much for your suggestions to note that the text is not sufficiently critical of the model results concerning the impact of the QBO on the polar vortex (Figs. 1 and 2). We revise the text to add these points and delete unnecessary descriptions.

The tracking change of old L251-255 is as follows:

"Most of the model correlations show smaller uncertainty than ERA5 due to having larger sample sizes. They have significant correlations over a range of altitudes only in a particular experiment. For example, at some altitudes GISS has a significant correlation in EN, MIROC-AGCM in LN, MRI-ESM2.0 in LN, WACCM in CTL, and MIROC-ESM has a significant correlation only in the lower stratosphere in CTL. LMDz has no or little correlations between the equatorial QBO winds and the polar vortex wind for any of the experiments. Models (ECHAM5sh, EMAC, EC-EARTH, MIROC-ESM, MRI-ESM2.0, and WACCM) have positive correlation profiles in ENSO-neutral, albeit weak compared to reanalysis. Most models do not show a significant correlation in EN, and only four models (MRI-ESM2.0, ECHAM5sh, EMAC, and MIROC-AGCM) out of 9 reproduce observed positive correlations with confidence intervals excluding zero at some altitudes. It is noted in Fig. 2 of Kawatani et al. (in revision) from their simple, time-height cross-section of the monthly and zonal-mean zonal winds over the equator in the EN and LN simulations that the QBO in the ECHAM5sh for the EN experiment is irregular, with stalling in downward phases of easterlies and westerlies. They showed that the QBOs in GISS and LMDz for the LN experiment are more irregular, and westerly phases sometimes fail to propagate into the lower stratosphere. These results ..."

**The tracking change of old L294-297 is as follows:**

"... Holton-Tan relationship in all three experiments (CTL, EN and LN). Only two of the models reproduce observed responses within a half of the amplitude for the ENSO-neutral case (MRI-ESM2.0 and WACCM), and only the MRI-ESM2.0 also shows a stronger impact on the QBO on the vortex under the LN condition than under EN condition (however, that model has the wrong sign response for EN). In LN, four models (ECHAM5sh, GISS, MIROC-AGCM, and MRI-ESM2.0) tend are better atte reproducinge the observed response, peaking at a slight amplitude of 3~36 m s-1 in the polar vortex region. Some models have significant composite differences of zonal wind in a particular experiment at 60° N and 10 hPa. For example, GISS and ECHAM5sh shows a significant difference in EN-peaking at 7 m s-1, and WACCM in CTL and a significant LN response just equatorward of 60 °N."

- The reviewer also asks for more clarification on the text on lines 221-226, and clarification on the statistical significance when multiple indices are used in the identification of the QBO.
- The analysis in the Walker Section initially used a consistent QBO definition and target season across all models. Specifically, we define the QBO using the zonal-mean zonal wind at 70 hPa during JJA. The corresponding figures are included in Fig. R2-3 of the responses to Reviewer #2 and supplementary material. With this uniform framework, we identify a coherent signal, but we want to enhance this signal and capture the strongest response in each model. To do so, we allow for slight adjustments in the QBO definition (70 or 85 hPa and JJA or SON) and in the target season (ranging from May to November), when necessary. We clarify this process in the revised text of Section 5.2. Importantly, when a model's QBO definition differs from the standard (70 hPa during JJA), we account for the increased flexibility by applying a Bonferroni correction. This reduces the significance threshold (alpha) from 0.05 to 0.025 or lower, depending on the number of alternative definitions tested. The significance threshold (also called the significance level) determines the p-value below which the null hypothesis is rejected. If the p-value is smaller than α, we reject the null hypothesis and consider the result statistically significant. We will clarify this point in the revised version of Section 2.
  - > Reviewer #1's suggestion "to include more in-text references to figure panels being discussed" would really help the reader.

Thank you very much for the suggestion from Reviewer #1 and the Editor. We revise the text to include more in-text references to figure panels being discussed.

Reviewer #2 also has excellent major comments and I strongly recommend you address them in your revised manuscript. In particular, Reviewer #2 asks for more discussion and analysis of why almost all the models do not reproduce three of the four teleconnections examined, and I agree. In some cases, further analysis may be required to support these discussions (e.g., is there a relationship between the model biases in the strength of the simulated QBO (in either neutral, El Nino and La Nina conditions) and the strength of the polar vortex response? Is there a relationship between the model biases in the strength of the polar vortex and the polar vortex response? Is there a relationship between biases in the extratropical stratospheric winds and the weakness in the impact of the QBO phase on the polar vortex?).

We appreciate your helpful suggestions. In the revised manuscript, we include more discussion and analysis of why almost all the models do not reproduce teleconnections examined. Please see our response to Reviewer #2 major comment R2-1 in more detail. Here, a summary of this discussion is as follows:

**QBOi ENSO experiments**

- ENSO modulation of the QBO in our QBOi ENSO experiments is investigated by a core paper of Kawatani et al. (in revision). https://egusphere.copernicus.org/preprints/2024/egusphere-2024-3270/
  - QBOs in some models are irregular, from a simple, time-height cross-section of the monthly and zonal winds in the El Nino and La Nina simulations, as shown in Figure 2 of Kawatani et al.
- 110 a) QBO teleconnections to polar vortex

- Problems of QBO teleconnection to the stratospheric polar vortex were investigated in detail by previous studies (Bushell et al., 2022; Anstey et al., 2022). As Anstey et al. (2022) described, the strength of the QBO teleconnection to the NH winter stratospheric polar vortex was shown to correlate with the amplitude of the QBO at 50 hPa. This altitude is the strongest correlation with the vortex in observations.
- Most models show poor performance of QBO amplitude at 50 hPa while climatological polar vortices in NH winter can be reproduced with their strength. These results are consistent with the hypothesis that unrealistically weak low-level QBO amplitude can weaken the teleconnection.
- 120 b) QBO teleconnections to subtropical jet
  - Models with larger QBO amplitudes do not necessarily exhibit stronger jet responses, nor do models with smaller amplitudes consistently show weaker responses. This means that neither the QBO amplitude nor the APJ position explains the inter-model spread in the QBO-APJ connection. Other factors may determine the QBO-APJ connection in the model.
- 125 c) QBO teleconnections to tropical precipitation
  - The combination of stratospheric and tropospheric biases in the tropics weakens the QBO signal reaching the tropical troposphere and contributes to inter-model differences in both the timing and spatial manifestation of the teleconnection.

Reviewer #2 also notes that previous work suggested that a measure of the efficacy of a model to reproduce QBO's impact on the polar vortex (the Holton-Tan effect) seen in observations is sensitive to the level that is used as an index of the QBO, and that model differences in the QBO justify the use of model-specific indices. Please address this point in your revised manuscript. Also, if you did choose levels to define the QBO that were model-specific, would the QBOs simulated by the models still be only half as strong as that observed (as documented in Fig. 3)? Would that still be the leading candidate for the weak relationships between the QBO phase, ENSO phase and polar vortex?

We appreciate your helpful suggestions. As we already described before, the problems of QBO teleconnection to the stratospheric polar vortex were investigated in the previous studies in detail. The QBO teleconnection to the NH winter stratospheric polar vortex is the strongest in observations when the QBO index is taken at 50 hPa (Anstey et al. 2022). Thus, we want to do model-observation comparison by applying the same QBO phase definitions to the models that are optimal for observed teleconnections, in order to determine if observed teleconnections are present in the model runs, without adjusting them on a model-by-model basis, for all analyses presented in this article.

In order to answer Reviewer #2 and Editor questions, we check levels to define the QBO that are based on observational studies (i.e., at 50 hPa) and that are based from a model specific level (i.e., at 30 hPa), as shown in Fig. R2-4 of the responses to Reviewer #2. Both panels (QBO50 and QBO30) show that most models underestimate QBOs and they are struggling to reproduce observed polar vortex responses to the QBO. Please see our response to Reviewer #2 major comment R2-4 in more detail.

Finally, Reviewer #2 asked why a different analysis procedure was used to examine the relationship between the QBO phase and the Walker circulation than that used to examine the other three teleconnections and whether the teleconnections were stronger for the Walker circulation simply because optimal pressure levels and seasons were chosen. I am not to bothered by this because, to be frank, the evidence presented in this section is pretty damning. Contrary to the description in the text, the observed relationship between the Walker circulation and the phase of the QBO shown is not well reproduced by most of the models for either La Nina or El Nino conditions. For La Nina conditions (Fig. 11), the anomalies in the zonal winds over the Pacific show a slightly westward shifted Walker circulation, whereas the models b,d,g,h and i shows a weakened Walker circulation (in phase anomalies of the opposite sign as the climatology aloft) and model e shows only easterly anomalies everywhere. The agreement during El Nino conditions is even worse (Fig. 12). [By the way, please note the contour interval for the anomalies in these figures. They seem to be much coarser than the discretized colorbars.] Stepping back a bit, I wonder whether the relationship between the QBO phase and the Walker circulation is poor because the band to define (5S-5N) the Walker circulation may be too narrow; 10S to 10N would better capture the zonal wind anomalies associated with the Walker circulation. Based on Fig. 17.17 of Wallace et al (2023), I expect this isn't the answer -- but it might be worth checking.

Thanks for your comments and suggestions. In response, we revise the analysis using the 10°S-10°N band, which better captures the zonal wind anomalies. The updated main figures now reflect this broader latitude band. Additionally, to provide more context and clarity, we include the results from our initial analysis, which focused on the target season JJA and used the standard QBO definition (zonal-mean zonal wind at 70 hPa during JJA) in the supplementary material. One of these figures (LN experiment) is presented in Figs. R1-1 (and R2-3; both figures are the same) of the responses to the reviewers. Such supplementary figures allow readers to better understand the progression of our approach. We also slightly adjust the main figures to align more closely with the standard QBO definition and the JJA season. The manuscript text will be revised accordingly to enhance clarity and ensure that the description of model performance is accurate.

**Minor comments:**

Does the GPCP bar in panel 9b stop the top of the plot, or does it run off scale? Why isn't there an error bar on this bar?

The GPCP bar runs off scale, so much so that even the error bar doesn't appear. The reason, to some extent, for this is the large signal due to the QBO-ENSO aliasing the manuscript discusses. We have produced two sets of figures for this plot, one where the y axis limits are set based on the GPCP bar and the other, like the original, where the limits are fixed to make the plot clearer. Both have positive and negative factors, and we provide the full figure in the revised Supplementary material of Fig. S8.

In Fig. 10, is the temperature also the zonal mean over the western Equatorial Pacific, or is it a zonal mean?

The temperature is also the mean over the western equatorial Pacific only. Thank you for the question; the revised manuscript clarifies this issue.

Lines 777-790: These statements are inconsistent with the published papers, dating back as far as Hoerling et al. (1987). Atmosphere general circulation models DO robustly reproduce the nonlinearity in the atmospheric response to warm and cold ENSO phases, given El Nino and La Nina SST anomalies.

Thank you for your suggestions. Our understanding is that observational evidence of mutual interactions between ENSO and QBO exists, but this possibility has not been widely studied using CMIP-class, climate model simulations. The observed ENSO-QBO relationship in current climate models is generally poorly reproduced, likely as a consequence of the coarse spatial resolution and the reliance on stationary parameterizations.

Serva, F., Cagnazzo, C., Christiansen, B., Yang, S.: The influence of ENSO events on the stratospheric QBO in a multi-model ensemble, Clim. Dyn., 54, 2561-2575, 2020, <a href="https://doi.org/10.1007/s00382-020-05131-7">https://doi.org/10.1007/s00382-020-05131-7</a>

Also, given the very weak relationship between the QBO phase, the ENSO phase, and the tropical anomalies shown in this study, it is unlikely that weaker ENSO events or ENSO events with less dramatic changes in the location of tropical convection than used in this study would yield further insights.

We agree with your second point that it is unlikely that weaker ENSO events or ENSO events with less dramatic changes in the location of tropical convection because of a weak relationship between the QBO phase, the ENSO phase, and the tropical anomalies are shown in this study. But, one study indicated that QBO is also influenced by the tropical SSTs in the Central Pacific (Shibata and Naoe, 2022), so that we believe that it will be worth further study of the role of ENSO flavors in the QBO-ENSO teleconnection.

Shibata K, Naoe H, 2022: Decadal amplitude modulations of the stratospheric quasi-biennial oscillation, J. Meteorol. Soc. Japan, 100, 29-44, https://doi.org/10.2151/jmsj.2022-001

**Response to Reviewer 1**

Title: QBOi El Niño Southern Oscillation experiments: Teleconnections of the QBO

Authors: Hiroaki Naoe, et al.

WCD manuscript on EGUsphere, MS No: egusphere-2025-1148

The authors would like to thank the reviewer for your time and effort in reviewing our manuscript entitled "QBOi El Nino Southern Oscillation experiments: Teleconnections of the QBO". Above all, the authors are deeply grateful for the many insights gained by reading the papers recommended by the reviewers. We will incorporate his/her valuable comments and suggestions on how our proposed revised manuscript will address your concerns. Our reviewer responses and revision plan are shown in blue text whereas reviewer's comments are shown in black. Individual responses are as follows.

Black: Reviewers comments

Blue: Authors response to the reviewer

**To Reviewer 1:**

RC1: 'Comment on egusphere-2025-1148', Anonymous Referee #1, 21 Apr 2025

This study uses ERA5 data and a multi-model ensemble of APARC QBOi models to investigate how QBO teleconnections are modulated by ENSO. To separate the QBO and ENSO signals, simulations were conducted with annually-repeating prescribed SSTs corresponding to idealized El Nino or La Nina conditions. Models are unable to represent the observed (ERA5) enhanced Holton-Tan effect during La Nina, where QBO W favors a stronger NH winter polar vortex. Models are also unable to represent the observed increase in SSWs during El Nino. Overall, the polar vortex responses to the QBO are much weaker than to ENSO in the models. In addition, the equatorward shift of the boreal winter Pacific subtropical jet (APJ) observed during QBO W in not seen in the models. In the tropics, the model experiments do not show a robust or coherent QBO influence on precipitation. It was further found that QBO effects on the Walker circulation exhibit a complex dependence on season, longitude, and phase of ENSO. They that suggested that weakness of the QBO polar vortex coupling in the models might arise from systematically weak QBO amplitudes at lower levels in the equatorial stratosphere, polar vortex biases in winter, and inadequate representation of stratospheric-troposphere coupling, while an inadequate representation of QBO effects in the tropical troposphere might arise from the systematically weak QBO amplitudes at lower levels, precipitation bias, and inadequate representation of the Walker circulation in these models. This paper documents the results of a considerable effort in the QBOi community, with well-organization presentation and choice of figures. The narrative provides an authoritative interpretation of the detail and status of observed and modeled QBO/ENSO influences on the extratropics. I recommend publishing with minor revision.

- R1-1. Idealized time mean La Nina and El Nino states. Would the model results be noticeably different for a time-varying ENSO (then binned by ENSO phase), versus two perpetual ENSO phases? It seems possible that the two-state method represents an upper bound on possible effects.
- In the context of tropical teleconnections, the two-state method versus a time-varying method might result in different model responses. In the tropics, a continuous ENSO state creates a different set of climatologies for the ITCZ, the Walker circulation, etc., which affect the way intraseasonal variability behaves in the model. Whether the two-state method is an upper bound on possible effects is less clear, since the evidence is not conclusive on why tropical precipitation responds to the QBO, but it is a fair hypothesis that needs to be tested.
- In the extratropics, QBO teleconnections are largely affected by tropical circulation and ENSO states, by means of subtropical jets, PNA pattern responses, stratospheric circulation including QBO itself, etc. Thus, the two-state method versus a time-varying method might result in different model responses in the extratropical teleconnections, too.
- R1-2. 1216-217, Fig. 13: This is a kind of discretized time-height section. It is similar to Reed et al.'s original 1961 figure which shows a time-height section of zonal wind. The Hovmoller diagram was originally defined to be the variation of geopotential height or another quantity near 60N as a function of longitude and time. It was generalized to mean a longitude-time diagram, which is usually used to indicate wave propagation. You have a table with dependence on season and altitude and you are not discussing wave propagation in longitude. Please use the phrase "season-altitude variation" instead of Hovmoller diagram to indicate what you are showing.

Thanks for this helpful clarification. Our analysis does not involve wave propagation in longitude. So, we revise the text accordingly without using "Hovmöller diagram".

L216-217 (old): "Hovmoller diagram" is deleted.

Fig. 13 caption: "Schematic Hovmoller diagram showing" is replaced with "Occurrence of"

R1-3. 1221-226: "when the QBO phase is not defined by the preferred 70 hPa level" does this mean that there are other ways to define it or that sometimes the 70 hPa level index isn't well defined? In this discussion of how multiple indices affect significance calculations, please give a sense of the meaning and outcome. For example, If you use more than one index definition at different levels, perhaps one might ascribe reduced significance to a result, but in your method it appears that alpha is reduced, therefore implying greater significance. A little more information would be helpful for understanding this paragraph.

Thank you for this thoughtful comment. The analysis in the Walker Section initially used a consistent QBO definition and target season across all models. Specifically, we define the QBO using the zonal-mean zonal wind at 70 hPa during JJA. Figure R1-1 (Fig. S10) shows the initial results for LN experiment and other figures are included in the revised supplementary material (Figs. S9, and S11) for consistency. With this uniform framework, we identify a coherent signal, but we want to enhance this signal and capture the strongest response in each model. To do so, we allow for slight adjustments in the QBO definition (70 or 85 hPa and JJA or SON) and in the target season (ranging from May to November), when necessary. These adjustments aim to capture the most robust response while maintaining a physically consistent framework. We have clarified this process in the revised text. Importantly, when a model's QBO definition differs from the standard (70 hPa during JJA), we account for the increased flexibility by applying a Bonferroni correction. This reduces the significance threshold (alpha) from 0.05 to 0.025 or lower, depending on the number of alternative definitions tested. The significance threshold (also called the significance level) determines the p-value below which the null hypothesis is rejected. If the p-value is smaller than α, we reject the null hypothesis and consider the result statistically significant. We clarify this point in the revised version of Section 2.

**The tracking change of old L221-226 is as follows:**

"ENSO composites in observations are done in the extratropics and subtropics for individual seasons (Sections 3, 4, and 5.2) and in the tropics for individual months (Section 5.1). In Section 5.2, tThe Bonferroni correction, as described by Holm (1979), is used for the two-sided *t*-test when the QBO phase is not defined by using the preferred 70 hPa level during June-July-August (JJA). In this method, the *p*-value-significance level of the statistical test is adjusted by dividing it by *m*, the number of tests performed, becoming more restrictive by increasing the confidence level. For instance, if the QBO definition is modified by season only, m = 2; if it is modified by both season and vertical level, m = 3. Accordingly,  $\alpha' = \alpha/m$ , where  $\alpha = 0.025$  (the 5% significance level for a two-sided test), and  $\alpha'$  denotes the adjusted *p*-valuethreshold-; implying that the corresponding p-value has to be smaller to reject the null hypothesis."

- Figure R1-1. Climatology (black contours) and QBO Westerly (W) minus Easterly (E) differences (shading and colored contours) in equatorial zonal wind profiles, averaged over 10° S-10° N, from the LN experiment for the QBO models. Black contours are drawn at 4 m s-1 intervals, and colored contour follow the same scale as the shading, as indicated in the color bar. The target season is JJA for all models, with the QBO phase defined at 70 hPa during JJA. Only statistically significant zonal wind differences at the 95% confidence level are shaded.
- 325 (Figures for CTL, LN, and EN experiments are added in the supplementary material of Figs. S9–11.)

- R1-4. Fig.1: It looks like only ECCAM5, WACCM and MRI are reasonably correct for neutral ENSO, but none get El Nino right. Maybe ECCAM5 and MRI get LaNina right (relative to ERA5).
- Thank you very much for your suggestions to note that the text is not sufficiently critical of the model results concerning the impact of the QBO on the polar vortex (Fig. 1). We revise the text to add these points and delete an unnecessary description.

  The tracking change of old L251-255 is as follows:
  - "Most of the model correlations show smaller uncertainty than ERA5 due to having larger sample sizes. They have significant correlations over a range of altitudes only in a particular experiment. For example, at some altitudes GISS has a significant correlation in EN, MIROC AGCM in LN, MRI ESM2.0 in LN, WACCM in CTL, and MIROC ESM has a significant correlation only in the lower stratosphere in CTL. LMDz has no or little correlations between the equatorial QBO winds and the polar vortex wind for any of the experiments. Models (ECHAM5sh, EMAC, EC-EARTH, MIROC-ESM, MRI-ESM2.0, and WACCM) have positive correlation profiles in ENSO-neutral, albeit weak compared to reanalysis. Most models do not show a significant correlation in EN, and only four models (MRI-ESM2.0, ECHAM5sh, EMAC, and MIROC-AGCM) out of 9 reproduce observed positive correlations with confidence intervals excluding zero at some altitudes."

R1-5. Fig. 2: Only MRI seems to represent the basic sense of the ERA5 signal.

Again, in the revised text, we add critical points of the model results and delete an unnecessary description.

The tracking change of old L294-300 is as follows:

"Only two of the models reproduce observed responses within a half of the amplitude for the ENSO-neutral case (MRI-ESM2.0 and WACCM), and only the MRI-ESM2.0 also shows a stronger impact on the QBO on the vortex under the LN condition than under EN condition (however, that model has the wrong sign response for EN). In LN, four models (ECHAM5sh, GISS, MIROC-AGCM, and MRI-ESM2.0) tend are better atto reproducinge the observed response, peaking at a slight amplitude of  $3\sim36$  m s-1 in the polar vortex region. Some models have significant composite differences of zonal wind in a particular experiment at 60° N and 10 hPa. For example, GISS and ECHAM5sh shows a significant difference in EN-peaking at 7 m s-1, and WACCM in CTL and a significant LN response just equatorward of 60 °N."

R1-6. Fig. 4 caption: suggest adding information to the effect of "La Nina, CTL, and El Nino, from left to right", to orient the reader about the order of the triplets, and maybe move to near the beginning of the caption.

Thank you very much for your suggestion. We move this description to near the beginning of the caption and change the order of the triplets as the reader is easy to identify them.

The tracking change of Fig. 4 caption is as follows:

- "Figure 4: SSW statistics ... daily data. The order of triplets from left to right are La Nina (LN, purple), ENSO neutral winter experiment (CTL, grey), and El Nino (EN, brown). Frequency ... ENSO neutral winter experiment (CTL, grey), El Niño (EN, brown), and La Niña (LN, purple)."
- R1-7. l356: suggest refer to (Fig. 4c). In this paragraph, and at times elsewhere, it might be beneficial to include more in-text references to figure panels being discussed.

Thank you very much for the suggestion from Reviewer #1 and the Editor. We revise the text to include more in-text references to figure panels being discussed.

**The tracking change of old L353-355 is as follows:**

- "Consistent with this expectation, in ERA5 during La Niña (the leftmost triplet of Fig. 4c), the final warming date in La Niña years is more delayed than that in ENSO-neutral and El Niño years. GISS and MRI-ESM2.0 exhibit later final warming dates in LN than in EN, which is similar to the observed response (Fig. 4c)."
  - R1-8. 1387, 150W-150E: How sensitive are results in Figs. 6 and 7 to the choice of longitude band?
- Thanks for the suggestion. We test the sensitivity of the QBO-APJ connection to the choice of longitudinal domain: (a) 150°E–150°W, as used in the original manuscript, (b) 130°E–120°W, as used in Anstey et al. (2022), and (c) 120°–180°E, as used in Park et al. (2022) (Figs. R1-2a–c, respectively). The domain adopted by Anstey et al. (2022) spans a broader longitudinal range than that used in this study, while the domain of Park et al. (2022) focuses on a region upstream of the jet core. Although a few models exhibit domain-dependent responses, the results are overall insensitive to the choice of longitudinal domain.
  - Anstey, J. A., Simpson, I. R., Richter, J. H., Naoe, H., Taguchi, M., Serva, F., ... & Yukimoto, S. (2022). Teleconnections of the quasi biennial oscillation in a multi model ensemble of QBO resolving models. Quarterly Journal of the Royal Meteorological Society, 148(744), 1568-1592.
- Park, C. H., Son, S. W., Lim, Y., & Choi, J. (2022). Quasi biennial oscillation related surface air temperature change over the western North Pacific in late winter. International Journal of Climatology, 42(8), 4351-4359.

Fig. R1-2. Sensitivity of the QBO-APJ connection to the longitudinal domain.

**Response to Reviewer 2**

Title: QBOi El Niño Southern Oscillation experiments: Teleconnections of the QBO

Authors: Hiroaki Naoe, et al. 395

WCD manuscript on EGUsphere, MS No: egusphere-2025-1148

The authors would like to thank the reviewer for your time and effort in reviewing our manuscript entitled "QBOi El Niño Southern Oscillation experiments: Teleconnections of the QBO". Above all, the authors are deeply grateful for the many insights gained by reading the papers recommended by the reviewers. We will incorporate your valuable comments and suggestions in our revised manuscript to address your concerns. Our reviewer responses and revision plan are shown in blue text whereas reviewer's comments are shown in black. Individual responses are as follows.

Black: Reviewers comments

Blue: Authors response to the reviewer

To Reviewer 2:

RC2: 'Comment on egusphere-2025-1148', Anonymous Referee #2, 01 May 2025 reply review of "QBOi El Nino Southern Oscillation experiments: Teleconnections of the QBO" by Naoe et al

This study aims to examine how QBO teleconnections are modulated by ENSO using a multi-model ensemble of QBOi models. The specific simulations examined are simulations in which SSTs are either climatological, El Nino, or La Nina, which allows for examining potential nonlinearities between QBO teleconnections and ENSO teleconnections. The use of ~10 models allows for assessment of model sensitivity and robustness. The authors examine four different QBO teleconnections - polar vortex response, subtropical jet, tropical precip, and Walker Cell. They conclude that the QBOi models generally fail to simulate the first three of these teleconnections, and hence it is difficult to conclude anything as to the possibility of ENSO and QBO teleconnections interacting. They find a robust effect of the QBO on the Walker Cell, however, the specifics of the QBO phase and season with maximum impact differ across the models.

While the paper should eventually be publishable in WCD, major revisions are needed first.

major comments:

R2-1. For the first three teleconnections where the models generally fail in the multi-model mean, there are still several models which are relatively more successful in capturing the observed response. There is no discussion of why there is spread across models for two of these teleconnections (vortex response and subtropical jet), while there is a very limited discussion of the third (namely Figure 10). The paper should include a detailed discussion for all three teleconnections as to possible causes of the intermodel spread in how well the models are doing. This could be similar to Figure 10, but instead of T100hPa, the authors could consider horizontal or vertical resolution, the mean state of the vortex or subtropical jet position, meridional width of the simulated QBO, strength of the QBO in each model in the lowermost stratosphere, strength of the QBO in the midstratosphere, etc. All of these factors could plausibly be linked to why some models are better than others, and by exploring all of them the paper might be able to provide some insights to model developers as to what needs to be improved.

We appreciate your helpful suggestions. In the revised manuscript, we include more discussion and analysis of why almost all the models do not reproduce teleconnections examined.

**a) Discussion for QBO teleconnections to polar vortex**

ENSO modulation of the QBO in the QBO is El Nino Southern Oscillation experiments is investigated by a core paper of Kawatani et al. (in revision; https://egusphere.copernicus.org/preprints/2024/egusphere-2024-3270/)

One of general characteristics of these experiments is that in the lower stratosphere, the westerly phase duration is generally longer in the La Nina simulations compared to the El Nino simulations. The downward propagation of QBO westerly and easterly phases to the lower stratosphere is more rapid during El Nino, which is a common characteristic among the models. However, QBOs in some models are irregular, from a simple, time-height cross-section of the monthly and zonal mean zonal winds over the equator in the El Nino and La Nina simulations for each model, as shown in Figure 2 of Kawatani et al. It is found that the QBO in the ECHAM El Nino experiment is irregular, with occasionally occurring in downward phases of easterlies and westerlies. The QBOs in GISS and LMDz La Nina experiments are more irregular, and westerly phases sometimes fail to propagate into the lower stratosphere.

Next, the QBO teleconnection problems relating to the stratospheric polar vortex teleconnection were investigated in more detail by previous studies (Bushell et al., 2022; Anstey et al., 2022). Figure 3 of Anstey et al. (2022) showed January correlation between vortex strength and equatorial wind at different altitudes for all models that performed CTL and for reanalyses. The strength of the QBO teleconnection to the NH winter stratospheric polar vortex was shown to correlate with the amplitude of the QBO at 50 hPa, which is the altitude that shows the strongest correlation with the vortex in observations. Most models show a statistically significant correlation at some altitudes, but the altitudes of peak correlation differ among models.

Figure 4 (b) of Anstey et al. (2022) showed January QBO-vortex correlation using 50 hPa QBO, versus QBO amplitude at 50 hPa. Models with weaker 50 hPa QBO amplitude show weaker correlation in January between the 50 hPa QBO wind and the polar vortex, consistent with the hypothesis that unrealistically weak low-level QBO amplitude can weaken the teleconnection.

In order to answer Reviewer #2 and Editor questions, we examine whether model performance of (a) QBO amplitude and/or (b) climatological polar night jet strength is related to the ability of model to capture the QBO-induced polar vortex responses (Fig. R2-1), assuming that the HTR relationship (polar vortex) route of the QBO teleconnection can be influenced by these two factors. The QBO amplitude is defined as the root mean square of the deseasonalized zonal wind time series at 50 hPa, multiplied by  $\sqrt{2}$ , following Dunkerton and Delisi (1985). QBO amplitudes at 50 hPa for most models are poor performance, in agreement with previous studies (Bushell et al., 2022; Anstey et al., 2022). It seems that larger QBO amplitudes at 50 hPa for models have larger polar vortex responses compared to the other models (but sometime wrong sign). Fig. R2-1(b) shows that climatological polar vortices in NH winter can be reproduced with observed strength. These results are consistent with the hypothesis that unrealistically weak low-level QBO amplitude can weaken the teleconnection.

These discussion points are added in Section 3.1.

Fig. R2-1. (a) Relationship between QBO amplitude at 50 hPa and composite difference of zonal-mean zonal wind (QBO50 W-E) at 60N and 10 hPa for CTL, El and LN experiments plus ERA5 (1959-2021) in units of m/s. The QBO definition index at 50 hPa is used. (b) Relationship between NH wintertime climatological zonal wind at 60N and 10 hPa and composite difference of zonal-mean zonal wind (QBO50 W-E) at 60N and 10 hPa for CTL, EN, and LN experiments including the ENSO neutral, El Nino, and La Nina winters for ERA5 in units of m/s.

**b) Discussion for QBO teleconnection to subtropical jet**

Given that the subtropical jet route of the QBO teleconnection can be influenced by (a) the QBO amplitude and/or (b) the climatological position of the subtropical jet (Garfinkel and Hartmann, 2011), we examine whether model performance in simulating these two factors is related to the ability of model to capture the QBO-induced shift of the Asian-Pacific jet (APJ) (Fig. R2-2). Here, the QBO amplitude is defined as the root mean square of the deseasonalized zonal wind time series at 70

hPa, multiplied by √2, following Dunkerton and Delisi (1985) and Bushell et al. (2022). The climatological position of the APJ is defined as the latitude of the maximum zonal-mean wind averaged over the APJ sector (150°E–150°W). Consistent with previous studies (Bushell et al., 2022; Anstey et al., 2022), most QBOi models underestimate the QBO amplitude. Only two models show a comparable QBO amplitude to the reanalysis. However, model biases in QBO amplitude do not affect those in the QBO-APJ connection (Fig. R2-2a). Models with larger QBO amplitudes do not necessarily exhibit stronger jet responses, nor do models with smaller amplitudes consistently show weaker responses. A similar result is also found in the APJ position (Fig. R2-2b). These results suggest that neither the QBO amplitude nor the APJ position explains the inter-model spread in the QBO-APJ connection. Other factors, such as transient and stationary eddies, may determine the QBO-APJ connection in the model. This possibility needs to be explored in a future study.

This discussion is added at the end of Section 4 and supplementary material of Fig. S3.

Fig. R2-2. Relationship between the QBO-induced APJ shift index and (a) QBO amplitude, and (b) subtropical jet latitude during ENSO-neutral (CTL) years.

**References:**

- 495 Garfinkel, C. I., and Hartmann, D. L.: The influence of the quasi-biennial oscillation on the troposphere in winter in a hierarchy of models. Part I: Simplified dry GCMs, J. Atmos. Sci., 68, 1273–1289, 2011.
  - Dunkerton, T.J. and Delisi, D.P. (1985) Climatology of the equatorial lower stratosphere. Journal of the Atmospheric Sciences, 42, 376-396.
- Bushell, A.C., Anstey, J. A., Butchart, N., Kawatani, Y., Osprey, S.M., Richter, J. H., 20 others: Evaluation of the Quasi-500 Biennial Oscillation in global climate models for the SPARC QBO-initiative, Q. J. Roy. Meteor. Soc., 148, 1459–1489, 2022. Anstey, J. A., Simpson, I. R., Richter, J. H., Naoe, H., Taguchi, M., Serva, F., 23 others: Teleconnections of the quasi-biennial oscillation in a multi-model ensemble of QBO-resolving models, Q. J. Roy. Meteor. Soc., 148, 1568–1592. https://doi.org/10.1002/qj.4048, 2022.

**c) Discussion for QBO teleconnection to tropical precipitation**

Several potential biases likely influence the tropical route of QBO teleconnections. Most proposed mechanisms linking the QBO to the tropical surface rely on interactions between the lowermost stratosphere and the uppermost troposphere. A key bias common to many models, including those used in this study, is a weak QBO amplitude in the lower stratosphere, which limits the effectiveness of stratosphere–troposphere coupling processes (Oueslati et al., 2013; Richter et al., 2020; García-510 Franco et al., 2023). Additionally, models exhibit persistent tropospheric biases, including the double Intertropical Convergence Zone (ITCZ) and unrealistic rainfall intensity distributions. These biases typically stem from model parameterizations, notably in convection and cloud microphysics schemes (Hagos et al., 2021). The combination of these stratospheric and tropospheric biases likely weakens the QBO signal reaching the tropical troposphere and contributes to intermodel differences in both the timing and spatial manifestation of the teleconnection. This helps explain why some models produce stronger signals during certain seasons or in particular regions compared to others.

This discussion is added at the end of Section 5.1.

- Hagos, S. M., Leung, L. R., Garuba, O. A., Demott, C., Harrop, B., Lu, J., & Ahn, M. S. (2021). The relationship between precipitation and precipitable water in CMIP6 simulations and implications for tropical climatology and change. *Journal of Climate*, *34*(5), 1587-1600.
- Richter, J. H., Anstey, J. A., Butchart, N., Kawatani, Y., Meehl, G. A., Osprey, S., & Simpson, I. R. (2020). Progress in simulating the quasi-biennial oscillation in CMIP models. *Journal of Geophysical Research: Atmospheres*, 125(8), e2019JD032362.
- Oueslati, B. and Bellon, G.: Convective entrainment and large-scale organization of tropical precipitation: Sensitivity of the CNRM-CM5 hierarchy of models, J. Climate, 26, 2931–2946, 2013.

On the topic of Figure 10, what is the correlation and slope of the best-fit line? Is the relationship statistically significant?

On the topic of Figure 10, the correlation of the best-fit line for all the data is -0.48 with a p-value of 0.02 according to a t-test of the Pearson correlation coefficient, indicating that the relationship is significant. However, these metrics are sensitive to the experiment, since a larger (in magnitude) correlation coefficient is found for El Niño conditions (r = -0.82) and a lower coefficient for La Niña experiments (r = -0.2).

R2-2. For the QBO signal in reanalysis, do you try to regress out a lingering signal of ENSO before plotting wqbo minus eqbo? Line 476-479 seems to indicate you don't do this, and it isn't clear whether this is done for the other teleconnections either. If this is not done, then comparing the observed signal to the model signal isn't a fair comparison as there will still be a residual signal from SSTs.

First, we do not consider ENSO to be a confounding factor in this study because our simulations explicitly isolate ENSO conditions. We know the idealized QBOi simulations cannot be directly compared to e.g. ERA5. Thus, this point is acknowledged in the revision of Section2:

"We note that the QBOiENSO experiments are idealized, therefore we mostly rely on observation-based datasets to determine whether the model responses are at least qualitatively in agreement with the (short) observational record."

We do not see clear advantages of regressing out an ENSO signal, compared to compositing on ENSO phase. In lines 476-479, for example, our composites of the observed precipitations (here GPCP, not a reanalysis) are made for El Nino and La Nina events. In this way, our comparison is 'apples-to-apples'. Regressing out an ENSO signal would make our analysis incomparable to our experiments.

R2-3. For the fourth teleconnection examined, the Walker Cell one, the authors adopt a completely different methodology than for the first three. Why for this section only do you play with the season and pressure level, but for earlier sections you don't? For the first three the models did a poor job, and now for this teleconnection they appear to be doing ok. Is this success for the Walker Cell just because you are giving the models lots of opportunities to succeed? Why not use this methodology for earlier sections too? Either way, the fact that a single paper is using very different methodological approaches for different sections is confusing, and leads to the (in my opinion misleading) impression that the models are much better at the QBO-> Walker Cell connection than the others.

Thanks for your comments and suggestions. To provide more context and clarity in the Walker circulation section, we also apply the same methodology used in the earlier sections, using a fixed QBO definition (in terms of season and vertical level) across all models. Our initial analysis reveals a coherent response in the Walker circulation compared to other sections. Figure R2-3 shows the initial results for LN experiment and other figures are included in the revised supplementary material for consistency (Figs. S9–11). Given this encouraging result, we explore whether the signal can be further strengthened by tailoring the QBO definition to each model, as a way to account for differences in how models represent the QBO vertical structure and timing.

Figure R2-3. Climatology (black contours) and QBO Westerly (W) minus Easterly (E) differences (shading and colored contours) in equatorial zonal wind profiles, averaged over 10° S-10° N, from the LN experiment for the QBOi models. Black contours are drawn at 4 m s-1 intervals, and colored contour follow the same scale as the shading, as indicated in the color bar. The target season is JJA for all models, with the QBO phase defined at 70 hPa during JJA. Only statistically significant zonal wind differences at the 95% confidence level are shaded.

This point is included at the beginning of Section 5.2:

"In this subsection, we examine whether a QBO impact on the Walker circulation can be detected across different ENSO phases. A recent study (Rodrigo et al., 2025) showed that in reanalyses the QBO signal in the divergent circulation is strongest over the Maritime Continent region in boreal summer (JJA), followed by autumn (SON), and weakest in winter. However, under El Niño and La Niña conditions this timing may slightly shift, potentially due to the ENSO influence on the QBO itself (Taguchi, 2010b; Kawatani et al., in revision). Additionally, model diversity and biases in the simulated QBO (Bushell et al., 2022) could lead to inter-model variations in the simulated QBO teleconnection. We begin the analysis by applying a common QBO definition and target season to all models, using the zonal-mean zonal wind at 70 hPa during JJA to define the QBO. With this approach, we identify a coherent signal, characterized by anomalous westerlies in the upper troposphere and anomalous easterlies in the lower troposphere over the Indian Ocean-Maritime Continent, in observations and some models (Figures S9, S10, and S11). To enhance this signal and capture To identify the strongest signal response in each model, we conduct a thorough search, allow slight adjustments todefining the QBO definition and target season when necessaryat slightly different seasons and vertical levels. The Bonferroni correction (Holm, 1979; see Section 2) is applied to the two-sided *t*-test when the QBO definition deviates from the preferred a level or season other than 70 hPa level during JJA is used to define the QBO phase."

One possibility of these more coherent responses is that the zonal circulation in these SST forced simulations is similar enough amongst models, due to the SST forcing, that the response is relatively more similar, whereas other aspects of the response, such as tropical precipitation, the polar vortex and the subtropical jet may be less constrained by the experimental setup. It is also plausible that the mechanisms that drive the Walker cell response are better represented in these models, given the relatively large static stability anomaly shown in the results, one could reasonably suspect that this mechanism could be large enough in the models to produce a consistent response.

R2-4. To be specific, previous work which allowed for different vertical levels to define the QBO can lead to very different conclusions as to whether models capture the HT effect of the polar vortex. See Rao et al 2020a. It could also be that the seasonality of the HT effect differs from one model to the next. It would be interesting to see if the QBO models still struggle to represent the HT effect if the authors adopted Rao et al's methodology.

As described in the method section, we do model-observation comparison by applying the same QBO phase definitions to the models that are optimal for observed teleconnections, in order to determine if observed teleconnections are manifested in the model runs. Thus, we use 'standard' indices (e.g., 50-hPa equatorial wind for the QBO), without adjusting them on a model-by-model basis, for all analyses presented in this article.

In Rao et al. (2020a), on the other hand, their QBO was defined at 30 hPa instead of 50 hPa, because some models largely underestimate the QBO magnitude in the lower stratosphere and for some models the QBO is difficult to detect at 50 hPa, and because the westerly phase lasts nearly as long as the easterly phase at 30 hPa.

These results suggest that our method is rather focuses on the identification of model biases by optimizing the observed teleconnections while Rao et al.'s method is to focus on the detection of model QBO signal, i.e., by maximizing the models' signals. In Garcia-Franco et al. (2022), when looking at Rao et al. (2020), which used QBO definitions at 30 hPa, they demonstrated that this level was not the most suitable for the tropical route.

An investigation of seasonality of Holton-Tan effect using different pressure levels would be a certainly interesting topic. But we think that only after identifying existing model biases that are done in the present work, we can move on to the next step, such a study of seasonality drift of Holton-Tan relationship using the phase-angle technique. Moreover, present-day simulations might be more appropriate to perform this kind of investigation, to better compare models and observation-based datasets.

Fig. R2-4. Relationship of composite difference of zonal-mean zonal wind between polar vortex responses at 60N and 10 hPa and QBO definition at 50 hPa (QBO50, left panel) and at 30 hPa (QBO30, right panel).

In order to answer Reviewer #2 and Editor questions, we check levels to define the QBO that are based on observational studies (i.e., at 50 hPa) and that are based on model specific level (i.e., at 30 hPa), as shown in Fig. R2-4. Both panels (QBO50 and QBO30) show that most models underestimate equatorial QBO composite differences at 50 hPa compared to those at 30 hPa and they are struggling to reproduce observed polar vortex responses to the QBO. For some models the QBO is difficult to detect at 50 hPa, similar to those described in Rao et al. (2020a), which was on CMIP models. In observations, a QBO response is large in La Nina in the left panel (QBO50) while a QBO response is largest in El Nino in the right panel (QBO30). As described in the Introduction, previous studies investigating the joint effects of QBO and ENSO on polar vortex

As described in the introduction, previous studies investigating the joint effects of QBO and ENSO on polar vortex variability in winter suggested that their interactions are nonlinear insofar as the Holton-Tan relationship is found to be significant in the La Nina phase but much weaker in the El Nino phase (Wei et al., 2007; Garfinkel and Hartmann, 2008; Calvo et al., 2009; Richter et al., 2011; Hansen et al., 2016). This means that our QBO-vortex responses in the observation classified in the ENSO phase using QBO50 is more consistent with previous studies.

This discussion is added in Section 3.1 and supplementary material of Fig. S1.

**minor comments:**

1. Somewhere in the paragraph from lines 109 to 116, and also near line 124, Rao et al 2020b should be cited and discussed Thank you for the helpful suggestion for this topic. We revise the text to cite this reference as they explored and evaluated three dynamical pathways for impacts of the QBO on the troposphere.

A discussion is added in the introduction around old L115:

"Also, Rao et al. (2020b) explored and evaluated three dynamical pathways (stratosphere polar vortex, North Pacific through the subtropical downward arching zonal wind, and tropical convection pathways) for impacts of the QBO on the troposphere, using the state-of-the-art CMIP5/6 models with a spontaneously generated QBO. They found that more than half of the models can reproduce at least one of the three pathways, but few models can reproduce all of the three routes."

**2. Line 135: Trascasa-Castro et al 2019 and Weinberger et al 2019 should be cited and discussed**

Thank you for the helpful suggestion. We revise the text to cite these references about the relationship between ENSO and SSWs.

A discussion is added in the introduction around old L135:

"For example, there is no indication of any nonlinearities between EN and LN, while SSW frequencies for EN and LN are both similar, using a chemistry-climate model (Weinberger et al., 2019). Trascasa-Castro et al. (2019) investigated the effect of variations in ENSO amplitude on European winter climate with idealized SST anomalies, and they did not find evidence of a saturation of the stratospheric pathway due to strong El Nino forcing, as suggested in previous literature."

**3. line 142-146: Ma et al should be cited and discussed**

Thank you for the helpful suggestion. We revise the text to cite this reference as QBO and ENSO have a nonlinear combined effect on North Atlantic surface pressure anomalies.

A discussion is added in the introduction around old L145-147:

"During El Niño, a stronger subtropical jet and the warmer polar vortex were present under QBO-W, while QBO anomalies in the tropical stratosphere were diminished and the poleward extent and amplitude of the QBO induced mean meridional circulation was reduced. Ma et al. (2023) assessed the synergistic effects of QBO and ENSO on the North Atlantic winter atmospheric circulation using model output and reanalysis data and found that the QBO and ENSO have a nonlinear combined effect on North Atlantic surface pressure anomalies, which arises because different pathways are preferred for different combinations of QBO and ENSO. In contrast, the polar vortex weakens ..."

4. line 199: Pahlavan et al should be cited.

This reference is included in the revised text.

technical edits aren't included in this round, but will be provided after the major comments are addressed.

---

## Editor Decision (ED1)

**Dear Prof. Naoe and co-authors:**

Thank you for your revised manuscript (WCD 2025-1148). I am recommending major revisions before the manuscript is accepted for publication, although they are better described as "minor but mandatory" revisions since they don't require much in the way of analysis. Rather, they are required because the discussion of the response of the Walker circulation should be revisited (see next paragraph), and the abstract and Short Summary still do not reflect the degree to which the models do not simulate the observed telelconnections between the QBO phase and the frequency of SWW in the polar vortex. Finally, I appreciate the additional discussion that offers reasons for why the models have such poor teleconnections to the polar vortex and the subtropical Pacific Jet. Your explanation for both is that the amplitude of the QBO in the lower stratosphere is too weak, and this is a very reasonable and important insight from the QBOi experiments. You should emphasize this in the discussion section and mention it in the abstract.

Please feel free to contact me if you have questions.

Regards, David

**Major comments:**

Concerning the Walker circulation, the original analysis used a season (JJA) and definition of the QBO that was based on observations and was applied to all models. The results showed little impact of the QBO phase on the Walker circulation (now in Figs. S9-S11). The revised manuscript uses time periods and QBO definitions that maximize the correlation between the QBO phase and the Walker circulation response in each model, and the results show that most models reproduce the observed circulation anomalies over the Indian Ocean and Maritime Continent. What is missing in this discussion is that, the difference in circulation over this region due to QBO-W minus QBO-E for the La Nina conditions is (given the weak statistical significance) basically the same for the El Nino conditions (c.f., Figs. 11 and 12) and (not surprisingly) for the control simulation (c.f., Fis S12 with Figs. 11 and 12). Hence, it should be noted in the text (and in the abstract) that the impact of QBO phase on the Walker circulation is insensitive to the phase of ENSO. Below, I also suggest a sentence to reflect this result be included in the abstract.

Abstract suggestions (note: all line numbers refer to the revised text, not the Author tracked changes).

• Lines 37-40: change to read "... are found in LN than in LN, although the differences in frequency are much smaller than that observed. Unlike in the observations, there

is no discernible difference in the QBO westerly (QBO-W) and QBO easterly (QBO-E) phases. The Asia-Pacific subtropical ...."

- Sentence starting on line 41 ("The sign and ... phases of ENSO"): delete this line because it is redundant with the sentence starting in the previous line ("IN the tropics...").
- Lines 45-46, modify to read "...and most models, with the QBO-W phase featuring upper-level westerly and lower-level easterly anomalies over the Indian Ocean-Maritime Continent relative to the QBO-E phase, although its amplitude and timing are model-dependent. In models, the impact of the QBO phase on the Walker circulation is insensitive to the phase of ENSO.

**Minor Comments**

The colorbar keys in Figure 2 are unreadable, and there are too many contours in the plots. Reduce the number of contours, or consider uneven contour intervals.

Figure caption 3: Delete the line "While for ... multi-model mean" and append this to the figure caption. "The dashed line in panel (a) shows the difference in observations when all years (1959-2022) are included in the analysis."

There is unnecessary and tedious detail in describing the deficiencies in nearly all the models to reproduce the observed relationship between the QBO phase on the phase of ENSO. Please delete lines 391-397 and replace them with the simple conclusion "Only one model (ECHAMsh) shows the observed relationship between the frequency of minor warmings and the phase of ENSO."

In Figure 4, please elaborate in the text why there are two (sometimes three) symbols for the same experiment in the GISS models and for the CNT experiment using the MIROC-ESM?

On Lines 540-541, "... and models show a distinct QBO signal between EN and LN experiments." I don't see this in Fig. 8. Figure 8 shows none of the models produce a robust precipitation response predicated on the phase of ENSO – anomalies are only a small fraction of those observed, and only (at best) a few percent of climatology.

Lines 612-614: That the observed precipitation response may depend on the sign of ENSO should be the first sentence in Section 5.1. The second line ("Overall, all ... experiments and models) is redundant with the discussion that immediately precedes it, and it should be deleted.

Figures 11 and 12. Again, the color of the contour lines does NOT match the shading scale in the colorbar; rather, the contour lines are all the same color. Please delete this phrase in both figure captions and indicate in the figure caption the contour interval of the plotted anomalies.

Line 717: ESM2.0 or MRI?

Lines 825-829: It should be noted that the impact of QBO phase on the Walker circulation is insensitive to the phase of ENSO.

---

## Author Response (AR2)

**Response to Editor**

Title: QBOi El Niño Southern Oscillation experiments: Teleconnections of the QBO

Authors: Hiroaki Naoe, et al.

WCD manuscript on EGUsphere, MS No: egusphere-2025-1148

The authors would like to thank the Editor for his time and effort in reviewing our manuscript entitled "QBOi El Niño Southern Oscillation experiments: Teleconnections of the QBO". Above all, the authors are deeply grateful for insights to reconsider us further to improve the paper. We incorporate your valuable comments and suggestions on our proposed revised manuscript. Our Editor's responses and revision are shown in "blue text", whereas Editor's comments are shown "in black". Individual responses to the Editor are as follows.

Black: Editor's comments

Blue: Authors response to the Editor, and all line numbers refer to the revised R1 text, not current revised R2 text.

To Editor:

Co-editor decision: Reconsider after major revisions by David Battisti, 17 July 2025

Dear Prof. Naoe and co-authors:

Thank you for your revised manuscript (WCD 2025-1148). I am recommending major revisions before the manuscript is accepted for publication, although they are better described as "minor but mandatory" revisions since they don't require much in the way of analysis. Rather, they are required because the discussion of the response of the Walker circulation should be revisited (see next paragraph), and the abstract and Short Summary still do not reflect the degree to which the models do not simulate the observed telelconnections between the QBO phase and the frequency of SWW in the polar vortex. Finally, I appreciate the additional discussion that others reasons for why the models have such poor teleconnections to the polar vortex and the subtropical Pacific Jet. Your explanation for both is that the amplitude of the QBO in the lower stratosphere is too weak, and this is a very reasonable and important insight from the QBOi experiments. You should emphasize this in the discussion section and mention it in the abstract.

Please feel free to contact me if you have questions.

- Thank you very much for your suggestions to note that the text is not sufficiently described about discussion of the Walker circulation and about the abstract and Short Summary which still do not reflect the degree to which the models do not simulate the observed teleconnections. We revised the text to add these points and reflect our model results that are struggling with the observed response in the abstract and Short Summary. Also, we emphasize important insight from the QBOi experiments that the most models are to struggle with observed responses in the polar vortex and subtropical jet due to weakened QBO amplitudes in the equatorial lower stratosphere.
  - the abstract and Short Summary still do not reflect the degree to which the models do not simulate the observed telelconnections between the QBO phase and the frequency of SWW in the polar vortex.
- $\rightarrow$  We reflect this point to add sentences in the Abstract:

L36-37: "... to idealized El Niño or La Niña conditions (QBOi EN and LN experiments, respectively). In LN, four out of nine models are to reproduce the observed Holton-Tan relationship within a half of the observed amplitude. In the Arctic winter climate, ...",

 $\rightarrow$  in the Short summary:

L51-52: "The polar vortex—QBO links While QBO teleconnections are qualitatively reproduced by the multi-model ensemble within a half of observed amplitude."

- → and in the discussion section:
- 55 L780: ", but much weaker than the observed response within at most a half of the observed amplitude (Fig. 2)."
- Finally, I appreciate the additional discussion that others reasons for why the models have such poor teleconnections to the polar vortex and the subtropical Pacific Jet. Your explanation for both is that the amplitude of the QBO in the lower stratosphere is too weak, and this is a very reasonable and important insight from the QBO experiments. You should emphasize this in the discussion section and mention it in the abstract.

Thank you very much for your suggestions. We emphasize important insight from the QBOi experiments in the discussion section that the most models are to struggle with QBO teleconnection in these regions and mention about that.

Abstract, L43-44: "... modulated by the prevailing phases of ENSO. Overall, the QBOi models show unrealistically weak QBO wind amplitudes in the lower stratosphere, which could explain the weak polar vortex and APJ responses and the weak precipitation signals in the tropics. The QBO teleconnection to the Walker circulation ... "

Short summary, L51-53: "While QBO teleconnections are qualitatively reproduced by the multi-model ensemble within a half of observed amplitude.", Poor performance of QBO signals in the tropics, subtropics, and polar regions is likelythey are not eonsistent due to unrealistically weak modelled QBO amplitudes in the lower stratosphere bias and other systematic model biases."

Discussion section, L783-784: "The observed strength ... the opposite direction (Fig. 3).

One may ask if a model-specific equatorial wind level such as 30 hPa can be more efficient for models to reproduce QBO's impact on the polar vortex (the Holton-Tan effect) than the standard 50-hPa equatorial wind that are optimal for observed teleconnections. However, for both 30-hPa and 50-hPa QBO indices most models underestimate equatorial QBOs and they are struggling to reproduce observed polar vortex responses to the QBO. We have examined whether model performance of QBO amplitude and/or climatological polar night jet strength is related to the ability of model to capture the QBO-induced polar vortex responses. QBO amplitudes at 50 hPa for most models are poor performance, while climatological polar vortices in NH winter can be reproduced with observed strength. This means that unrealistically weak low-level QBO amplitudes can weaken the QBO teleconnections to the polar vortex, as indicated by the previous QBOi multi-model ensemble studies (Richter et al., 2022; Anstey et al., 2022c).

Major SSWs occur ... "

Discussion section, L798: "The APJ changes ... capture a stronger APJ in EN than in LN. We have also examined whether the subtropical jet route of the QBO teleconnection can be influenced by the QBO amplitude and/or the climatological position of the subtropical jet. Although most QBOi models underestimate the QBO amplitude, models with larger QBO amplitudes do not necessarily exhibit stronger jet responses, nor do models with smaller amplitudes consistently show weaker responses. This means that neither the QBO amplitude nor the APJ position explains the inter-model spread in the QBO-APJ connection. Other factors, such as transient and stationary eddies, may determine the QBO-APJ connection in the model.

The positive equatorial Pacific signal ... "

Major comments:

Concerning the Walker circulation, the original analysis used a season (JJA) and definition of the QBO that was based on observations and was applied to all models. The results showed little impact of the QBO phase on the Walker circulation (now in Figs. S9-S11). The revised manuscript uses time periods and QBO definitions that maximize the correlation between the QBO phase and the Walker circulation response in each model, and the results show that most models reproduce the observed circulation anomalies over the Indian Ocean and Maritime Continent. What is missing in this discussion is that, the difference in circulation over this region due to QBO-W minus QBO-E for the La Nina conditions is (given the weak statistical significance) basically the same for the El Nino conditions (c.f., Figs. 11 and 12) and (not surprisingly) for the control simulation (c.f., Fis S12 with Figs. 11 and 12). Hence, it should be noted in the text (and in the abstract) that the impact of QBO phase on the Walker circulation is insensitive to the phase of ENSO. Below, I also suggest a sentence to reflect this result be included in the abstract.

I agree on the missing discussion about the Walker circulation. According to your suggestion, we add a sentence in the abstract: L45-46: "... and most models, with the QBO-W phase featuring upper-level westerly and lower-level easterly anomalies over the Indian Ocean–Maritime Continent relative to the QBO-E phase, although its amplitude and timing are model-dependent. In models, tThise impact of the QBO phase on the Walker circulation appears not to be is insensitive to the phase of ENSO."

And, we modify the sentence in the discussion section:

L825-827: "A distinct QBO signal, characterized by upper-level westerly and lower-level easterly anomalies, is observed in the equatorial troposphere in ERA5, which is not very sensitive to the ENSO phase during both El Niño and La Niña years."

Abstract suggestions (note: all line numbers refer to the revised text, not the Author tracked changes).

- Lines 37-40: change to read "... are found in LN than in LN, although the differences in frequency are much smaller than 120 that observed. Unlike in the observations, there is no discernible difference in the QBO westerly (QBO-W) and QBO easterly (QBOE) phases. The Asia-Pacific subtropical ...."

According to your suggestion, this part is modified to:

L37-39: "... are found in EN than LN73 and unlike the observations The frequencythere is no discernible differences of SSW frequencies between QBO westerly (QBO-W) and QBO easterly (QBO-E) phases are indistinguishable, suggesting that the polar vortex responses to the QBO are much weaker than those to the ENSO in these models."

- Sentence starting on line 41 ("The sign and ... phases of ENSO"): delete this line because it is redundant with the sentence starting in the previous line ("IN the tropics ...").

We agree on redundance with these sentences, and we merge them together into as follows:

L41-43: "In the tropics, these experiments do not show a robust or coherent QBO influence on precipitation. The sign and spatial pattern of the QBO precipitation response vary widely across models and experiments, indicating that any potential QBO signal is strongly modulated by the prevailing phases of ENSO."

- Lines 45-46, modify to read "...and most models, with the QBO-W phase featuring upper-level westerly and lower-level easterly anomalies over the Indian Ocean-Maritime Continent relative to the QBO-E phase, although its amplitude and timing are model-dependent. In models, the impact of the QBO phase on the Walker circulation is insensitive to the phase of ENSO. According to your suggestion, this part is modified to:

L45-46: "... and most models, with the QBO-W phase featuring upper-level westerly and lower-level easterly anomalies over the Indian Ocean–Maritime Continent relative to the QBO-E phase, although its amplitude and timing are model-dependent. In models, tThise impact of the QBO phase on the Walker circulation appears not to beis insensitive to the phase of ENSO."

**Minor Comments**

The colorbar keys in Figure 2 are unreadable, and there are too many contours in the plots. Reduce the number of contours, or consider uneven contour intervals.

- → The colorbar keys in Figure 2 are revised to be readable.
- → We consider uneven contour intervals in the tropics and extratropics:

L333: "Contour intervals are is 3 (10) m s-1 north (south) of 20° N."

Figure caption 3: Delete the line "While for ... multi-model mean" and append this to the figure caption. "The dashed line in panel (a) shows the difference in observations when all years (1959-2022) are included in the analysis."

→ This part of Figure caption 3 is revised as follows:

L356-357: "... in each QBO phase. The dashed line in panel (a) shows the QBO composite difference in ERA5 While for the experiments, ENSO is neutral, when all years (1959–2022) in ERA5 are included in the analysis (1959–2022). MMM ..."

There is unnecessary and tedious detail in describing the deficiencies in nearly all the models to reproduce the observed relationship between the QBO phase on the phase of ENSO. Please delete lines 391-397 and replace them with the simple conclusion "Only one model (ECHAMsh) shows the observed relationship between the frequency of minor warmings and the phase of ENSO."

→ We agree with you and delete these unnecessary and tedious details. We modify these sentences as follows:

L390-397: "... not capture the LN response (e.g., EC-EARTH, MIROC-AGCM, MRI-ESM2.0). Only one model (ECHAMsh) shows the observed relationship between the frequency of minor warmings and the phase of ENSO, and it ECHAMsh has similar frequencies in CTL and LN and more events in EN. GISS shows large spreads in CTL and EN, suggesting that the response is not statistically robust. In Fig. 4b, frequencies of minor SSWs are similar in both ENSO-neutral and El Niño years and there are fewer events in La Niña years in ERA5. There is a large spread in minor SSW frequencies between the models. EC-EARTH and ECHAM5sh show high frequencies of minor SSWs in EN whereas LMDz and MRI-ESM2.0 show low frequencies of minor SSWs. MIROC-AGCM produces fewer SSWs in the CTL, EN, and LN experiments, and MIROC-ESM shows relatively higher frequencies for both major and minor SSWs in EN and LN compared to the other MIROC model. The GISS ensemble shows large spread in all three experiments, suggesting an important role for internal variability."

In Figure 4, please elaborate in the text why there are two (sometimes three) symbols for the same experiment in the GISS models and for the CNT experiment using the MIROCESM?

→ These are ensemble members. Some models in CTL, EN, and LN experiments have two-to-three ensemble simulations and other models have only one member. No daily zonal wind data were archived for WACCM, for the rest the plot follows the availability reported in Table 1. We update an explanation in the figure caption 4 as follows:

L403: "..., occurring across full seasons. It is noted that Ddifferent multiple marker signs in the same experiment of a triplet for a modelare used to indicate ensemble members (depending on data availability).; and uUncertainties are estimated ..."

On Lines 540-541, "... and models show a distinct QBO signal between EN and LN experiments." I don't see this in Fig. 8. Figure 8 shows none of the models produce a robust precipitation response predicated on the phase of ENSO – anomalies are only a small fraction of those observed, and only (at best) a few percent of climatology.

- → We agree, the meaning of the sentence was to convey to the reader that in some models, the precipitation or OLR signals are different between experiments. For example, EC-EARTH shows a significant positive signal in the equatorial western Pacific in the LN experiment, which is the opposite sign in the EN experiment. Similarly, in LMDz this same region, including Australia, shows opposing signs of the signal. These anomalies are higher than 1.4 mm day in magnitude, so we couldn't really say that they are only a few percent of the climatology, we think they're large enough to be worth mentioning.
- → This sentence is modified as follows:

L540-541: "None of the models show an OLR signal comparable to observations, and some models (EC-EARTH, ECHAM5sh, LMDz and GISS) show OLR (and precipitation; Fig. 8) responses that appear distinct QBO signal between EN and LN experiments in regions such as the equatorial Pacific. In other words, there is no consistent or robust precipitation response across models or experiments."

Lines 612-614: That the observed precipitation response may depend on the sign of ENSO should be the first sentence in Section 5.1. The second line ("Overall, all ... experiments and models) is redundant with the discussion that immediately precedes it, and it should be deleted.

- → We agree. These 2 sentences are deleted. Instead, they are placed to the beginning of Section 5.1 with some wordings.
- → L522: "5.1 Tropical precipitation

Several studies have proposed that the observed signal from the QBO on tropical precipitation depends on the underlying ENSO phase (e.g., Taguchi et al., 2010, García Franco et al., 2022, 2023). This section investigates this hypothesis using these QBOi models and experiments through the analysis of tropical precipitation and OLR. Figure 8 ..."

→ L611-615: "These results emphasize that the QBO signal on tropical precipitation may strongly depend on the state of ENSO as suggested by the observations (García-Franco et al., 2023). Overall, some models show a robust and significant precipitation response to the QBO but this response is distinct from observations and varies in sign and magnitude amongst experiments and models."

Figures 11 and 12. Again, the color of the contour lines does NOT match the shading scale in the colorbar; rather, the contour lines are all the same color. Please delete this phrase in both figure captions and indicate in the figure caption the contour interval of the plotted anomalies.

 $\rightarrow$  We agree. The figure 11 caption is modified as follows:

L701: "Black contours are drawn at 4 m s-1 intervals. Colored contours use the same intervals as the shading, with red contours indicating positive and blue indicating negative differences. and the colored contour intervals match the shading seale in the color bar. The target season ..."

Line 717: ESM2.0 or MRI?

→ MRI-ESM2.0 is right.

Lines 825-829: It should be noted that the impact of QBO phase on the Walker circulation is insensitive to the phase of ENSO.

We modify it as follows:

L825-827: "A distinct QBO signal, characterized by upper-level westerly and lower-level easterly anomalies, is observed in the equatorial troposphere in ERA5, which is not very sensitive to the ENSO phase during both El Niño and La Niña years."

---

## Editor Decision (ED2)

**Dear Prof. Naoe and co-authors:**

Thank you for your third revision on the manuscript (WCD 2025-1148). I appreciate that you and your co-authors have corrected the grammar and presented a manuscript that is now clear of ambiguities due to language problems. In doing so, I was able to find a few minor (but important issues that should be easy to address quickly. I am recommending minor revisions to allow you to make these modifications (or to argue why certain modifications are not appropriate).

Please feel free to contact me if you have questions.

Regards, David

Note: all line numbers refer to the revised text, not the Author tracked changes. Text appearing in the revised manuscript is in *italics*; my suggested changes are in red.

- As stated on lines 303-304, "no model reproduces the observed strength of the
  Holton-Tan relationship between the phase of the QBO and the strength of the polar
  vortex across all three experiments". That is true, but I don't think it goes far enough.
  Following that sentence, I suggest you insert the following sentence: "From Fig. 2,
  only three models reproduce the observed relationship in the CTL and EN
  experiments, and only one model reproduces the observed relationship in the LN
  experiment."
- Lines 305-306: the correlation between U60 and Ueq50 in LN observations is positive (and statistically different from zero), while in the GISS model LN experiment is zero, so I don't think it is fair to say this is a good fit compared to observations. So I would remove GISS from the list on line 306 and change to read "In LN three models..."
- Change lines 38-39 in the Abstract to reflect the above results. Change "... respectively). In LN, four out of nine models reproduce the observed Holton–Tan relationship within half of the observed amplitude." to read "... respectively). In LN, four out of nine models reproduce the observed Holton–Tan relationship within half of the observed amplitude." to read " The strength of the Holton–Tan relationship between the phase of the QBO and the strength of the polar vortex seen in observations is reproduced in fewer than three models under ENSO neutral conditions and by one model under EN conditions. In LN, three out of nine models reproduce the observed Holton–Tan relationship, but with less than half of the observed amplitude."

- Line 241, change to read "... of the statistical test  $\alpha$  is adjusted ..."
- Fig. 1, titles on panels c, f and j should probably read CTL49EN39LN39, CTL69EN81LN81, and CTL87EN99LN99, respectively.
- The sentence on lines 354-356 (In summary, across all three experiments, models generally show.... observed response.) is still ambiguous. If this statement is meant to say "In summary, for each experiment, models generally show ...", then it isn't supported by Figs. 2 (or 3): for the EN experiment, Fig. 2, five of nine models produce the opposite response to that in observations. I would change this sentence to read "In summary, across the CNT and LN experiments, models generally show.... observed response.)"
- Line 412, change to read "... also show later final warming dates in LN ..." to read "... also show median final warming dates that are later in LN ..."
- Line 414, change "fail to show earlier final warming dates" to read "show later final warming dates"
- Line 445, change "... on the APJ. Only ..." to read "... on the Asian-Pacific subtropical jet (APJ). Only ..." so the reader doesn't have to search 10 pages back to recall what APJ means.
- Line 548, change to read "... precipitation response to the phase of the QBO across models or experiments."
- Line 642, change to read "... expect models that have larger TTL ..."
- Line 650, change to read ".. underestimate QBO TTL temperature anomalies ..." because zonal wind, in itself, isn't the zero-order control on convection.
- Lines 735-736, change "...exhibit a tropospheric signal characterized by upper-level (100 hPa) westerly and lower-level (700 hPa) easterly anomalies during various seasons from May to November. This pattern suggests ..." to read "...exhibit a tropospheric signal characterized by upper-level (100 hPa) westerly anomalies during various seasons from June to November, while about half of the models and

the observations show lower level (700 hPa) easterly anomalies. This pattern suggests ..."

- Delete the two sentences on lines 783-785 ("The observed .... the observations.") because it is irrelevant to what follows in this paragraph.
- Lines 808-809, change to read "... show weaker responses. Hence, neither the ..."
- Line 811, change to read "The impact of the QBO on the troposphere is examined, focusing ...."
- Line 812, change to read "... to the QBO phase in the ..."
- Line 820, change to read "... is strongest in observations over the..."
- Line 829-830. I am not sure what is being argued here. Yes, the SST forcing (common to all the models) constrains the circulation in the equatorial (lon-height) plane, and so each model has a very similar mean state that is being acted upon by the QBO. Hence, one might expect the QBO W minus QBO E response to look the same across the models for the same experiment. But the SST for the LN experiment is different from the SST for the EN experiment, and so the mean state that QBO acts upon in the LN experiment is different from that in the EN experiment. And yet, the QBO W minus QBO E response is very similar in the LN and EL experiments (compare the colored contours in Fig. 11 to those in Fig. 12). This implies that the impact of the QBO phase is not terribly sensitive to changes in the SST. Also, on Line 831-2, it is stated that precipitation may be less constrained by the experimental setup but we know that tropical Pacific precipitation is heavily constrained by the SST, which is common to all models for each experiment. Altogether, I find this paragraph confusing or even troublesome, and so I would drop it.
- Line 858, change "... vortex coupling arise from consistently weak QBO amplitudes at lower levels in the equatorial stratosphere biases..." to read "vortex coupling arises from consistently weak QBO amplitudes at lower levels in the equatorial stratosphere, biases..."

---

## Author Response (AR3)

**Response to Editor**

Title: QBOi El Niño-Southern Oscillation experiments: Teleconnections of the QBO

Authors: Hiroaki Naoe, et al.

WCD manuscript on EGUsphere, MS No: egusphere-2025-1148

The authors appreciate the Editor's extensive efforts in reviewing our manuscript. As senior author I appreciate the chance in

this round of revision to improve the overall standard writing. Indeed, a careful rereading of the previous version revealed a

number of grammatical errors and sentences that seem obscure in English.

Therefore, in this revision I have improved this manuscript in two aspects:

- Consistency of writing style

- Improvement of the level of English

A fundamental revision in this MS was made in the "Discussion and Summary" Section. Here, there was a lack of consistency

in the discussion of the different aspects of our analyses. First, I completely revised many problematic sentences. Then, I

rewrote the manuscript trying to maintain the consistency of my writing style. I also revised problematic sentences in other

sections and in the figure captions. Next, I improved the English of this revised manuscript using the AI program ChatGPT in

addition to English-proof online tools such as Grammarly, Ginger, Wordvice, trinka. (Note that I have included an

acknowledgement of the use of ChatGPT.)

Once it was somewhat complete, I had the co-authors check it over and received more input about grammatical errors and

other problematic sentences.

I believe that the revised manuscript addresses the quality of writing issue that raised by the Editor and hope that this manuscript

will be acceptable for publication in WCD.

(Blue: Author's response to the Editor)

1

**30 Black: Editor's comments**

Co-editor decision: Reconsider after major revisions by David Battisti, 07 August 2025

Public justification (visible to the public if the article is accepted and published):

Dear Prof. Naoe,

The scientific content of this paper is now fine, but the quality of the presentation is not. While I understand that English is not the first language of many of your co-authors, the quality of the writing in the revised manuscript is not acceptable for publication in WCD. Grammatical errors are frequent, and they render the content of many of the sentences obscure. Please work with your co-authors who have a command of English in order to craft a manuscript that will not overly burden and frustrate the reader. I am confident that revisions that address this issue will result in a paper that will then be accepted for publication in WCD.

Cheers, David

---

## Author Response (AR4)

**Response to Editor**

Title: QBOi El Niño-Southern Oscillation experiments: Teleconnections of the QBO

5 Authors: Hiroaki Naoe, et al.

WCD manuscript on EGUsphere, MS No: egusphere-2025-1148

The authors appreciate the Editor's extensive efforts in reviewing our manuscript. We incorporate your valuable comments and suggestions on our revised manuscript. Our Editor's responses and revision are shown in "blue text", whereas Editor's comments and suggested changes are shown "in black" and "in bold red", respectively. Individual responses to the Editor are as follows. (All line numbers refer to the third revision, not current fourth revision.)

Co-editor decision: Publish subject to minor revisions (review by editor), 24 Sep 2025

by David BattistiSupplement to the public justification (visible to the public if the article is accepted and published) (pdf):

15 egusphere-2025-1148-comments-to-author.pdf

Public justification (visible to the public if the article is accepted and published):

Dear Prof. Naoe and co-authors:

Thank you for your third revision on the manuscript (WCD 2025-1148). I appreciate that you and your co-authors have corrected the grammar and presented a manuscript that is now clear of ambiguities due to language problems. In doing so, I was able to find a few minor (but important issues that should be easy to address quickly. I am recommending minor revisions to allow you to make these modifications (or to argue why certain modifications are not appropriate).

Please feel free to contact me if you have questions.

Regards, David

25 → We appreciate the Editor's positive assessment of our revised manuscript. We carefully considered each of the remaining comments and revised the manuscript where appropriate.

Note: all line numbers refer to the revised text, not the Author tracked changes. Text appearing in the revised manuscript is in italics; my suggested changes are in red.

- As stated on lines 303-304, "no model reproduces the observed strength of the Holton-Tan relationship between the phase of the QBO and the strength of the polar vortex across all three experiments". That is true, but I don't think it goes far enough. Following that sentence, I suggest you insert the following sentence: "From Fig. 2, only three models reproduce the observed relationship in the CTL and EN experiments, and only one model reproduces the observed relationship in the LN experiment."

→ We insert this sentence as follows:

L304: "... across all three experiments (CTL, EN, and LN). From Figure 2, only three models reproduce the observed relationship in CTL and EN, and only one model (MRI-ESM2.0) reproduces the observed relationship in LN. Only The two models in CTL (MRI-ESM2.0 and WACCM) exhibit responses within half the observed amplitude in CTL. The one model Furthermore, only in LN (MRI-ESM2.0) exhibits a stronger QBO impact on the vortex in LN..."

- Lines 305-306: the correlation between U60 and Ueq50 in LN observations is positive (and statistically different from zero),
  45 while in the GISS model LN experiment is zero, so I don't think it is fair to say this is a good fit compared to observations. So
  I would remove GISS from the list on line 306 and change to read "In LN three models...."
  - → We revise the sentence as follows:

L304: "... In LN, threefour models (ECHAM5sh, GISS, MIROC-AGCM, and MRI-ESM2.0) better reproduce the observed response, ..."

50 }

- Change lines 38-39 in the Abstract to reflect the above results. Change "... respectively). In LN, four out of nine models reproduce the observed Holton-Tan relationship within half of the observed amplitude." to read "... respectively). In LN, four out of nine models reproduce the observed Holton-Tan relationship within half of the observed amplitude." to read " The strength of the Holton-Tan relationship between the phase of the QBO and the strength of the polar vortex seen in observations is reproduced in fewer than three models under ENSO neutral conditions and by one model under EN conditions. In LN, three out of nine models reproduce the observed Holton-Tan relationship, but with less than half of the observed amplitude."
- → We reflect this point to add sentences in the Abstract:
- L38-39: "... representing idealized El Niño and La Niña conditions (the QBOi EN and LN experiments, respectively), and results are compared with the QBOi control experiment (CTL) under ENSO-neutral conditions. The strength of the Holton-Tan relationship between the phase of the QBO and the strength of the polar vortex seen in observations is reproduced in fewer

than three models in CTL and by one model in EN. In LN, three four out of nine models reproduce the observed Holton–Tan relationship, but withwithin less than half of the observed amplitude."

65

- Line 241, change to read "... of the statistical test a is adjusted ..."
- → We insert this symbol as follows:
- L241: "In this method, the significance level of the statistical test  $\alpha$  is adjusted by dividing it by m, ..."

70

- Fig. 1, titles on panels c, f and j should probably read CTL49EN39LN39, CTL69EN81LN81, and CTL87EN99LN99, respectively.
- → You are right. Something was missing in the titles on these panels. We revise this figure.

75

- The sentence on lines 354-356 (In summary, **across** all three experiments, models generally show.... observed response.) is still ambiguous. If this statement is meant to say "In summary, for **each** experiment, models generally show ...", then it isn't supported by Figs. 2 (or 3): for the EN experiment, Fig. 2, five of nine models produce the opposite response to that in observations. I would change this sentence to read "In summary, **across the CNT and LN experiments**, models generally show.... observed response.)"
- → We revise the sentence as you suggested:
- L354-356: "In summary, across the CTL and LNall three experiments, models generally show ...the observed response."

- Line 412, change to read "... also show later final warming dates in LN ..." to read "... also show median final warming dates that are later in LN ..."
- → We revise the sentence as you suggested:
- L412: "GISS and MRI-ESM2.0 also show median later final warming dates that are later in LN compared with EN, consistent with the observed response (Fig. 4c)."
  - Line 414, change "fail to show earlier final warming dates" to read "show later final warming dates"

- → We revise the sentence as you suggested:
- 95 L414: "In contrast, five models (EC-EARTH, ECHAM5sh, LMDZ, MIROC-AGCM, and MIROC-ESM) fail to show laterearlier final warming dates in EN, opposite to the observed response."
- Line 445, change "... on the APJ. Only ..." to read "... on the **Asian-Pacific subtropical jet (APJ)**. Only ..." so the reader doesn't have to search 10 pages back to recall what APJ means.
  - → We spell out APJ here as follows:
  - L445: "... by ENSO, focusing on the Asia-Pacific subtropical jet (APJ). Only ..."
- Line 548, change to read "... precipitation response to the phase of the QBO across models or experiments."
  - → We revise the sentence as you suggested:
  - L304: "In summary, there is no robust or consistent precipitation response to the phase of the QBO across models or experiments."
  - Line 642, change to read "... expect models that have larger TTL ..."
  - → We revise the sentence as you suggested:

110

- L642: "One might expect models that havewith larger TTL temperature signals or static stability to also show stronger precipitation signals."
- Line 650, change to read ".. underestimate **QBO TTL** temperature anomalies ..." because zonal wind, in itself, isn't the zero-order control on convection.
- → We revise the sentence as you suggested:
- 120 L649-651: "Overall, the QBOi models underestimate QBO wind amplitudes in the lower stratosphere (Bushell et al., 2022) and thus show weak. TTL temperature anomalies (Serva et al., 2022), which may explain their weak precipitation signals."

- Lines 735-736, change "...exhibit a tropospheric signal characterized by upper-level (100 hPa) westerly and lower-level (700 hPa) easterly anomalies during various seasons from May to November. This pattern suggests ..." to read "...exhibit a tropospheric signal characterized by upper-level (100 hPa) westerly anomalies during various seasons from June to November, while about half of the models and the observations show lower level (700 hPa) easterly anomalies. This pattern suggests ..."
  - → We revise the sentence as you suggested:
- 130 L734-736: "Across all three experiments, nearly all models, along with ERA5, exhibit a tropospheric signal characterized by upper-level (100 hPa) westerly and lower-level (700 hPa) easterly anomalies during various seasons from May to November, while about half of the models and the observations show lower level (700 hPa) easterly anomalies. This pattern suggests ..."
- Delete the two sentences on lines 783-785 ("The observed .... the observations.") because it is irrelevant to what follows in this paragraph.
  - → We delete these sentences as you suggested:

L783-786: "The observed correlation coefficients between the 50 hPa equatorial zonal wind and the strength of the polar vortex at stratospheric altitudes during DJF exhibit considerable uncertainty (Fig. 1a). The models show less uncertainty because of their larger sample sizes (Fig. 1). Some models reproduce weaker correlations for a specific ENSO experiment, consistent with the observations. The Holton–Tan relationship ..."

- Lines 808-809, change to read "... show weaker responses. Hence, neither the ..."
- 145  $\rightarrow$  We revise the sentence as you suggested:

L808-809: "... show weaker responses. Hence, This means that neither the QBO amplitude ... "

- Line 811, change to read "The impact of the QBO on the troposphere is examined,
- 150 focusing ...."

140

→ We revise the sentence as you suggested:

L811: "The impacttropical pathway of the QBO teleconnection modulated by ENSOon the troposphere is examined, focusing on tropical precipitation (Figs. 8–10) and the Walker circulation (Figs. 11–13). "

- Line 812, change to read "... to the QBO phase in the ..."
- → We revise the sentence as you suggested:
- L815: "The precipitation response to the QBO phase in these experiments varies by model, region, and ENSO phase, ..."

160

- Line 820, change to read "... is strongest in **observations** over the..."
- → We revise the sentence as you suggested:

L304: "The QBO teleconnection to the Walker circulation is strongest in reanalysesobservations over the Indian Ocean–Maritime Continent region ..."

165

175

- Line 829-830. I am not sure what is being argued here. Yes, the SST forcing (common to all the models) constrains the circulation in the equatorial (lon-height) plane, and so each model has a very similar mean state that is being acted upon by the QBO. Hence, one might expect the QBO W minus QBO E response to look the same across the models for the same experiment. But the SST for the LN experiment is different from the SST for the EN experiment, and so the mean state that QBO acts upon in the LN experiment is different from that in the EN experiment. And yet, the QBO W minus QBO E response is very similar in the LN and EL experiments (compare the colored contours in Fig. 11 to those in Fig. 12). This implies that the impact of the QBO phase is not terribly sensitive to changes in the SST. Also, on Line 831-2, it is stated that precipitation may be less constrained by the experimental setup but we know that tropical Pacific precipitation is heavily constrained by the SST, which is common to all models for each experiment. Altogether, I find this paragraph confusing or even troublesome, and so I would drop it.
- → We delete this paragraph as you suggested:

L304: "One possible explanation for the more coherent Walker circulation response is that the zonal circulation in the SST forced simulations is sufficiently similar across models—owing to the SST forcing—that the responses remain relatively consistent. In contrast, other aspects of the response, such as the tropical precipitation, the polar vortex, and the subtropical jet, may be less constrained by the experimental setup. It is also plausible that the mechanisms driving the Walker cell response are better represented in these models. Given the relatively large static stability anomaly shown in the results (Fig. 10), one could reasonably suspect that this mechanism is strong enough in the models to produce a consistent response in the Walker circulation."

185

- Line 858, change "... vortex coupling arise from consistently weak QBO amplitudes at lower levels in the equatorial stratosphere biases..." to read "vortex coupling arises from consistently weak QBO amplitudes at lower levels in the equatorial stratosphere, biases..."
- → We revise the sentence as you suggested:
- L857-860: "In the extratropical stratosphere, previous studies using QBOi models have suggested that the systematic weakness of the QBO-polar vortex coupling arises from consistently weak QBO amplitudes at lower levels in the equatorial stratosphere biases in the wintertime polar vortex, and inadequate representation of stratosphere-troposphere coupling (Bushell et al., 2022; Richter et al., 2022; Anstey et al., 2022c). "